



**In-depth study of the formation processes of single atmospheric particles in**
**the southeastern margin of Tibetan Plateau**
Li Li[1,3], Qiyuan Wang[1,2], Jie Tian[1], Huikun Liu[1], Yong Zhang[1], Steven Sai Hang Ho[4],
Weikang Ran[1], Junji Cao[5]
[1] Key Laboratory of Aerosol Chemistry and Physics, State Key Laboratory of Loess and Quaternary
Geology, Institute of Earth Environment, Chinese Academy of Sciences, Xi'an 710061, China
[2] CAS Center for Excellence in Quaternary Science and Global Change, Xi'an 710061, China
[3] University of Chinese Academy of Sciences, Beijing 100049, China
[4] Division of Atmospheric Sciences, Desert Research Institute, Reno, NV 89512, United States
[5] Institute of Atmospheric Physics, Chinese Academy of Sciences, Beijing 100029, China
Correspondence to: Qiyuan Wang (wangqy@ieecas.cn) and Junji Cao (jjcao@mail.iap.ac.cn).



**Abstract**

The unique geographical location of the Tibetan Plateau (TP) plays an important role in regulating global climate change, but the impacts of the chemical components and atmospheric processing on the size distribution and mixing state of individual particles are rarely explored in the southeastern margin of the TP, which is a transport channel for pollutants from Southeast Asia during the pre-monsoon season. Thus a single-particle aerosol mass spectrometer (SPAMS) was deployed to investigate how the local emissions of chemical composition interact with the transporting particles and assess the mixing state of different particle types and secondary formation in this study. The TP particles were classified into six main types: the rich-potassium (rich-K) type was the largest fraction of the total particles (30.9%), followed by the biomass burning (BB) type (18.7%). Most particle types were mainly transported from the surroundings and cross-border of northern Myanmar; but the air masses from northeastern India and Myanmar show a greater impact on the number fraction of BB (31.7%) and Dust (18.2%) types, respectively. Besides, the two episodes events with high particle concentrations showed that the differences in the meteorological conditions in the same air clusters could cause significant changes in chemical components, especially the Dust and EC-aged types changed by a sum of 93.6% and 72.0% respectively. Ammonium and Dust particles distribute at a relatively larger size ($\sim$ 600 nm), but the size peak of other types is present at $\sim$ 440 nm. The easily volatilized nitrate ($^{62}NO_3^-$) during the transport process leads the more abundant sulfate ($^{97}HSO_4^-$) to mix internally with the TP particles. $C_2H_3O^+$, $HC_2O_4^-$, $NH_4^+$, $NO_3^-$, and $HSO_4^-$, severed as the indicators of secondary formation, are present in the atmospheric aging process of photo-oxidation and aqueous reaction by a linear correlation with $O_x$ ($O_3$+$NO_2$) and relative humidity (RH). This study provides insights that can improve the knowledge of particle composition and size, mixing state, and aging mechanism at high time resolution over the TP region.

**Keywords**

Southeastern Tibetan Plateau, Pre-monsoon, Individual particles, Chemical characteristics, Atmospheric aging,



## 1   Introduction

Atmospheric aerosols have complex components and sources and can be coated with inorganic or organic materials during transport and atmospheric processing (Crippa et al., 2013). When further coated through coagulation, condensation, and photochemical oxidation, their physical and chemical properties and optical properties will change greatly, making their impact on the air more uncertain (Jacobson, 2002; Zaveri et al., 2010; Matsui, 2016; Budisulistiorini et al., 2017). The ability of aerosol particles to affect atmospheric conditions is dependent on their sizes, chemical compositions, and mixing states (Mc Figgans et al., 2006; Dusek et al., 2006; Ma et al., 2012). For example, dust aerosol is an important factor affecting climate change through the interactions of various physical processes such as the direct radiative effect (Mahowald et al., 2014; Shao et al., 2011; Wang et al., 2022); the formation of secondary chemical components often aggravates regional pollution. However, the ability of secondary formation has regional differences due to the variations in precursors, source strengths, and meteorological conditions (Pratt et al., 2011; Volkamer et al., 2006). The influence of these complex chemical components on aerosol size and mixing state varies with the pollution sources and/or atmospheric formation mechanism, which have been widely studied in urban of low altitude (Pratt et al., 2011; Liu et al., 2020a; Xu et al., 2017). Furthermore, a previous study found that the migration or formation of low-volatile component (such as nitrate and organic matter) could effectively be reduced due to evaporation during the upward transportation process (Liu et al., 2020b), which further alter the chemical compositions and the particle sizes. In addition, the transportation of the aerosols to a relatively cleaner environment prevails the formation of secondary chemicals at a high altitude (Liu et al., 2020b). Therefore, a comprehensive investigation of the detailed characteristic of aerosol formation and mixing states is required to understand their environmental effects in low-, and high-altitude.

As a typical high-altitude region, the Tibetan Plateau (TP) has the highest and largest mountain area in the world, which is the most sensitive and obvious indicator of climate change in the entire Asian continent (Yao, et al., 2012; Chen and Bordoni, 2014; Immerzeel et al., 2010). In recent decades, many studies have shown that the melting and retreat of glaciers



in the TP regions is accelerating, and the main reason is attributable to anthropogenic
emissions, such as greenhouse gases and aerosols (William et al., 2010; Luo et al., 2020; Hua
et al., 2019). Atmospheric aerosols also can influence the properties and life span of clouds as
cloud condensation nuclei, and affect local hydrological cycles and monsoon patterns by
changing the microphysical properties of clouds (Qian et al., 2011; Seinfeld and Pandis, 2012,
Xia et al., 2007; Gettelman et al., 2013; Kumar et al., 2017). The southern part of the TP is
always affected by the transport of more polluted air from South Asia along the mountain
valleys, especially during the pre-monsoon (i.e., March-May) with the southwest prevailing
wind (Cao et al., 2010; Chan et al., 2017; Zhu et al., 2017; Zhao et al., 2017; Han et al., 2020).
Most studies have focused on the influence of optical properties; however, a few studies have
been conducted on aerosol components within the plateau.

Present aerosol components studies conducted in TP mostly focus on exploring the

influence of light-absorbing carbon aerosols and dust particles on climate change by optical
or offline sampling methods (e.g., Wang et al., 2019a; Liu et al., 2021). There is a lack of
studies on the full composition, mixing states, and formation mechanism of aerosols in the
southeast margin and even the entire TP, especially using high-time resolved measurements.
Although time-integrated sampling with filter collection followed by laboratory analyses has
been widely adopted for the chemical characterization of aerosols (Chen et al., 2015; Li et al.,
2022a; Shen et al., 2015; Zhang et al., 2013). Drawbacks of the traditional approach include
low time resolution, high detection limit, and time- and labor-intensive procedures. More
advanced aerosol online measurement devices with a high temporal resolution, such as the
aerosol chemical speciation monitor (ACSM) and aerosol mass spectrometer (AMS) (Ng et
al., 2011; Canagaratna et al., 2007) show the inability to measure the mixing state of particles
and limit to obtain the information of non-refractory submicron aerosol. AMS/ACSM mainly
used to provide online observation datasets with high temporal resolution (including the mass
concentration of sulfate, nitrate, ammonium, chloride, and organic; and corresponding mass
spectral), which is beneficial to obtain the dynamic processes of source emission in the
atmosphere (Du et al., 2015; Zhang et al., 2019a). At the same time, aerosol time-of-flight
mass spectrometry (ATOFMS) (Prather et al., 1994), and single particle aerosol mass
spectrometer (SPAMS) (Li et al., 2011), are popular for characterizing atmospheric individual



particles. These devices could determine the full chemical composition and their size
distribution, to achieve more detailed information, such as the dynamic processes of chemical
aging, mixing state, and transport of the aerosols (Wang et al., 2016; Zhang et al., 2015,
2016). To the best knowledge, the advanced measurement device has not yet been applied for
the studies conducted in TP, leading to a lack of in-depth research on the $PM_{2.5}$ pollution in
TP, especially in the southeastern margin. The shortage of information hinders our
understanding of the distribution characteristics and formation mechanism of full aerosol
components in high-altitude regions.

The southeastern margin of the TP is an important transitional zone between the

high-altitude TP and the low-altitude Yungui Plateau (Wang et al., 2019a; Zhao et al., 2017),
an ideal place for investigating the impacts of pollutants transport and formation in the
high-altitude zone. In this study, a high-time resolution field observation of individual
particles (SPAMS) was deployed on the southeastern margin of the TP during the
pre-monsoon, to continuously (i) investigate the changes of chemical characteristics between
transport and local fine particles during pre-monsoon, (ii) determine the particle size
distributions, and the mixing states of different particle types, and (iii) assess the
contributions of photooxidation and aqueous reaction to the formation of the secondary
species. These results would expand our understanding of the chemical components, size
distribution, mixing state, and aging pathways of aerosols in the high-altitude areas in the TP
and surrounding areas.
**2    Methodology**
**2.1 Observation site**

Intensive 1-month field observation was deployed at the rooftop ($\sim$ 10 m above the ground)

of the Lijiang Astronomical Station, Chinese Academy of Sciences (3260 m above sea level;
26°41′24″N, 100°10′48″E), Gaomeigu County, Yunnan Province, China, during the
pre-monsoon period (from April 14th to May 13th, 2018). The nearest residential area is the
Gaomeigu village (3–5 km away) with a small population size of 113 residents in 27
households. The villagers make a living by farming (e.g., potato and autumn rape), and
biomass is the main residential fuel (Li et al., 2016). The site is surrounded by rural and


mountainous areas and has no obvious industry or traffic emissions. During the observation
period, the average temperature (T) and relative humidity (RH) are 8.4 ± 3.1°C and 69% ±
21%, respectively. The wind speed (WS) is 2.2 ± 1.2 m·s$^{-1}$ with the prevailing wind in the
north and northeastern (Fig. S1).

**2.2 On-line instrument**

A detailed operational principle and the calibrations of the single-particle aerosol mass
spectrometer (SPAMS, Hexin Analytical Instrument Co., Ltd., Guangzhou, China) has been
described elsewhere (Li et al., 2011). Briefly, individual particles are drawn into SPAMS
through a critical orifice. The particles are focused and accelerated, then aerodynamically
sized by two continuous diode Nd: YAG laser beams (532 nm), subsequently desorbed and
ionized by a pulsed laser (266 nm) triggered exactly based on the velocity of the specific
particle. The generated of positive and negative molecular fragments are recorded with the
corresponding size of individual particles. In summary, a velocity, a detection moment, and
an ion mass spectrum are recorded for each ionized particle, while there is no mass spectrum
for not ionized particles. The velocity could be converted to $d_{va}$ based on a calibration using
polystyrene latex spheres (PSL, Thermo Scientific Corp., Palo Alto, USA) with predefined
sizes. The average ambient pressure is 690 hPa (in a range of 685–694 hPa) during the
measurements and calibration. Particles measured by SPAMS mostly are within the size
range of vacuum aerodynamic diameter ($d_{va}$) 0.2–2.0 μm.
Meteorological parameters, including the planetary boundary layer (PBL), temperature
(°C), RH (%), WS (m·s$^{-1}$), and wind direction (WD) were continuously achieved using an
automatic weather station (Model MAWS201, Vaisala HydroMet, Helsinki, Finland) at a time
resolution of 5 min. Gaseous concentrations (ppbv) were obtained using a multiple gas
analyzer (Thermo Scientific Corp.), including ozone (O$_3$, model 49i) and nitrogen oxides
(NO$_x$, model 42i) in a 5-min resolution. The SPAMS and gas analyzers are co-located in the
same position, while the weather station was uncovered outside ~5 m from the sampling
house. Time series of SPAMS particles, gaseous concentrations (NO, NO$_x$, O$_3$, and CO) and
meteorological parameters (PBL, temperature, RH, WD, and WS) were shown in Fig. S2.

**2.3 Individual particle classification**



During the observation period, a total of 461,876 ambient particles with the size ($d_{va}$) of
0.2–2.0 μm were collected, including 55,583 in Episode I (E1; from April 18[th] 08:00 to April
19[th] 08:00) and 62,110 in Episode II (E2; from April 26[th] 17:00 to April 28[th] 02:00). The
analyzed particles are classified into 1,557 clusters using an adaptive resonance theory neural
network (ART-2a) with a vigilance factor of 0.8, a learning rate of 0.05, and 20 iterations
(Song et al., 1999). Finally, eight major particle clusters [i.e., Potassium-rich (rich-K),
Biomass Burning (BB), Organic Carbon (OC), Ammonium, Aging Element Carbon
(EC-aged), Dust, Sodium, Potassium-containing (NaK-SN), and Iron (Fe)-Lead
(Pb)-containing (Metal)] with distinct chemical patterns were manually combined, which
represent ~99.7 % of the population of the detected particles. The remaining particles are
grouped as "Other". The characteristics of the positive and negative mass spectra (MS) of
each particle type are shown in Fig. S3. A detailed description of classification criteria for
individual particles and the characteristic ion fragments for each particle type can be found in
Text S1.
**2.4 Trajectory-related analysis**
To determine the influence of regional transport on different particles at Gaomeigu, the
trajectory clusters analysis was carried out using the 72-h backward air mass trajectories at
500 m above the ground level. The trajectories were calculated with the Hybrid
Single-Particle Lagrangian Integrated Trajectory model (Draxler and Hess, 1998), and the
meteorological data were obtained from the Global Data Assimilation System (GDAS;
ftp://arlftp.arlhq.noaa.gov/pub/archives/gdas1, last access: 6 April, 2022). The cluster analysis
employs a Euclidean-oriented distance definition to differentiate and cluster the major spatial
features of the inputting trajectories. Details of the trajectory clustering method can be found
in Sirois and Bottenheim (1995). To investigate the effects of transport on the chemical
characteristic of the individual particles, trajectories with particle number concentrations high
than the 75[th] percentile are considered as pollution (Liu et al., 2021).
**3   Results and Discussion**
**3.1 Characteristics of particle composition**



Table 1 summarizes the numbers of concentrations, relative percentages, and

characteristic ions of each particle type. The most dominant particle type in Gaomeigu during
pre-monsoon is rich-K, accounting for an average of 30.9% of the total resolved particles,
followed by BB (18.7%), OC (12.8%), Ammonium (11.9%), EC-aged (10.9%), and Dust
(10.7%). Similar to the results of some studies in urban areas, rich-K or
carbonaceous-containing type is the dominant particle type (15-50%) (Li et al., 2014; Zhang
et al., 2015; Shen et al., 2017; Zhang et al., 2017; Xu et al., 2018). Differently, few
researchers can capture the high proportion of Ammonium particles as shown in this study
(Shen et al., 2017; Xu et al., 2018), which is ascribed to the conversion of ammonia ($NH_3$)
precursor emitted from large-scale agricultural activities and mountain forest (Engling et al.,
2011; Li et al., 2013). It is necessary to point out that 60% of Ammonium particles contain
signals of diethylamine (DEA, $^{58}C_2H_5NHCH_2^+$), implying their similar formation pathway
(Zhang et al., 2012). Moreover, the DEA-containing particle represented 12.5% of the total
ambient particles, which is significantly higher than that in some urban areas at low altitudes
(around 2%) (Cahi et al., 2012; Pratt and Prather, 2010; Zhang et al., 2015; Li et al., 2017)
but is comparable to the observed in processing of high RH, fog and cloud events at a high
altitude (> 9%) (Roth et al., 2016; Lin et al., 2019). This suggests that the formation of
amines under high RH and fog condition might exist in the Gaomeigu area (with an altitude
of 3260 m), for example the high relative fraction of DEA-containing particle corresponds to
a high RH (Fig. S4), and the existence of amine sources govern the ammonium formations
(Bi et al., 2016; Rehbein et al., 2011; Zhang et al., 2012). The relatively larger fraction of
Dust particles is related to the short-time occurrences of dust events in spring (Fig. S5),
leading to a wide contribution ranging between 10% and 70% in the period of 19:00 on April
16th to 10:00 on April 17th.

Fig. 1 shows the diurnal variations of each particle type. The rich-K, BB, and OC

particles decrease after midnight until 06:00, possibly explained by the curtailment of local
traffic and biomass-burning activities even though both the planetary boundary layer (PBL)
height and WS decrease (Fig. S6). Then, they rapidly increase around 07:00 when the
pollutants from biomass burning are transported from the upwind region as the PBL rises
(Liu et al., 2021). At 11:00, the particle counts sharply decrease till 16:00–17:00, caused by



the pollutant dispersion with the increases of the PBL height and WS. Increasing trends are
observed after 17:00 due to the reduction of PBL height and WS. In contrast, the Ammonium,
EC-aged, and Dust particles show a unimodal pattern of the daily diurnal variation (Fig. 1d–f).
From 00:00 to 06:00, minor fluctuation of particle concentrations of Ammonium, EC-aged,
and Dust is observed for these particle types. After that, they continuously elevate until 12:00
due to the regional transport, traffic emission, and road dust from upwind areas (Text S2).
While the PBL height and WS increase continuously, the Ammonium, EC-aged, and Dust
particles begin to decline from 12:00 to 17:00. The subsequent increases of these three
particles after 17:00 are attributed to the reduction of PBL height, as a result of the
accumulation of pollutants in the near-surface atmosphere.

Based on the transport pathways, four air masses clusters are identified to investigate the

effect of regional transport on the major particle types (i.e., rich-K, BB, OC, Ammonium,
EC-aged, and Dust) (Fig. 2). The most dominant air masses are Cluster 1, 3 and 4 from
northeastern Myanmar, accounting for 59.8%, 33.2% and 4.6% of the total trajectories,
respectively. Cluster 1 had an average percentage of 31.2%, 20.2%, 13.7%, 11.8%, 10.9%,
and 6.5%, respectively, on the rich-K, BB, OC, Ammonium, EC-aged, and Dust particles.
Clusters 3 and 4 had comparable contributions of BB, OC, Ammonium, and EC-aged to those
of Cluster 1, but with a high contribution of Dust, which approximately 12.2% and 18.2% of
Clusters 3 and 4 are referred to as dust pollution. The diurnal variations of the BB and OC
fractions are similar which rapidly elevate at 07:00 (Fig. S7) due to the increased contribution
of biomass burning and vehicle emissions from Cluster 1, Ammonium and EC-aged particles
(peak at 07:00) caused by the effect of Cluster 1 and 3 together. The similar diurnal trend of
Clusters 3 and 4 are both associated with dust contributions, which decrease at 04:00 and
increase at noon. The increased nighttime particles could be attributed to the pollutant
accumulation with the decreased PBL height. Cluster 2 originated from northeastern India
and passes over Bangladesh. This cluster accounts for only 2.4 % of the total trajectories, in
which ~30.1% and ~31.7% are mainly associated with the rich-K and BB particles,
respectively. Even though Clusters 2 and 4 are composed of a small fraction of total
trajectories (2.4% and 4.6 %, respectively), BB and dust particles are identified as the major





pollutants, suggesting significant influences from India and northeastern Myanmar during the
campaign.

### 3.2 Characteristics of size distribution and mixing state

Fig. 3 shows the size distributions of each particle type. Corresponding the average MS
(Text S1 and Fig. S2), rich-K, BB, and OC, EC-aged particles have similar sources from
vehicle emission or solid-fuel combustion, their size distribution presents at small-scale
(~440 nm) (Fig. 3a). However, the relative proportion of each particle type is distinct under
different sizes range, maybe due to the different atmospheric processing in ambient. For
example, as shown in Fig. 3b, the percentage of rich-K increases from 17% to 44% along
with the increase of particle size from 200 to 420 nm, and then decreases to <10% at 900 nm;
a similar thing, the percentage of BB increases from 9% to 27% with the increasing sizes of
200 to 420 nm, and then decreases to <10% at 660 nm. However, the OC and EC-aged types
are mainly distributed in relatively small sizes, the percentage of them gradually decreases
from 31% and 36% to 9% when size range from ~200 to 500 and ~400 nm, respectively.
Notably, the Ammonium and Dust are mainly distributed in large sizes of ~600 nm (Fig. 3a).
The Ammonium particles gradually increase from 1.6% to 29% from 440 to 740 nm, then
decline to <10% at 1.2 μm. The relatively large size distribution of the Ammonium type is
ascribed to the intense atmospheric aging during regional transport (Text S1). The percentage
of Dust particles gradually increases from 10% at 560 nm to 60% at 1.48 μm. This is
consistent with consist to the fact that the dust is a coarse particle, generally formed at the
roadside and fly ash.
To investigate the mixing state of the secondary species in the six main particle types,
several number fractions of secondary markers (i.e., $^{97}HSO_4^-$, $^{195}H(HSO_4)_2^-$, $^{62}NO_3^-$, $^{18}NH_4^+$,
$^{58}C_2H_5NHCH_2^+$ and $^{89}HC_2O_4^-$) are selected (Fig. 4). Amine particles are characterized by ion
signals of amine at $m/z$ $^{58}C_2H_5NHCH_2^+$ (diethylamine, DEA) (Angelino et al., 2001; Moffet et
al., 2008) and sulfuric acid at $m/z$ $^{195}H(HSO_4)_2^-$, which is indicative of acidic particles
(Rehbein et al., 2011).
The most abundant number fraction of $^{97}HSO_4^-$ and $^{18}NH_4^+$ in Ammonium (99% and
94%) and EC-aged (92% and 31%) particles, but the proportion of $^{62}NO_3^-$ is lower (2% and



7%), suggest that ammonium sulfate is not a predominant form instead of ammonium nitrate. Meanwhile, a high number fraction of $^{195}H(HSO_4)_2^-$ and DEA is also observed in Ammonium (63% and 60%) and EC-aged (4% and 19%) particles. These abundant mixtures may represent the high hygroscopicity of Ammonium and EC-aged particles, and their ability to neutralize the acidic particles of Ammonium particle (Sorooshian et al., 2007). Then, a moderate proportion of $^{97}HSO_4^-$ and $^{18}NH_4^+$ are seen on the rich-K (65%, 7%) and OC (56%, 4%) particles. In contrast, more $^{62}NO_3^-$ contributes to the rich-K (38%) and OC (68%) particles, mainly affected by vehicle emissions (Text S1). Followed by BB (18%) and Dust (6%) particles are found a low number fraction of $^{97}HSO_4^-$, the moderate $^{62}NO_3^-$ accounts for 45% of the BB particle and only 3% of the Dust particle, and $^{18}NH_4^+$ is minor (＜1%), which is suggested the degree of BB and Dust particles aging are low. In addition, representing the component of secondary organic formation, oxalate ($^{89}HC_2O_4^-$) is mainly mixed with BB (13%) and rich-K (12%) particles, because they have similar sources from biomass burning. A relatively low fraction (<5%) of the oxalate-containing particles in the OC, Ammonium, EC-aged, and Dust particles are a result of their source origins.

Our results are inconsistent with the observations in other field studies (Zhang et al., 2015; Dall'Osto and Harrison, 2012; Li et al., 2022b), in which the carbonaceous-containing particles are more mixed with sulfate than nitrate, additionally, the rich-K and Dust particles are inclined to mix with nitrate. The dissimilarity could be potentially ascribed to the high emission of sulfate from coal combustion, biomass burning, and vehicles to the rich-K, Ammonium, and EC-aged particles (Yang et al., 2017; Li et al., 2022b), and the higher loss of more volatile nitrate than sulfate during the airmass transportation.

**3.3 Formation process of the high number concentration particle episodes**

A more in-depth investigation of the characteristics of the main particle types in the southeastern Tibet Plateau was conducted during two episode periods when the number concentration of particles was high (i.e., E1: from 08:00 April 18th to 08:00 April 19th, 2018; E2: 17:00 April 26th to 02:00 April 28th, 2018). Even though the two episode events are mostly contributed by Cluster 1, the chemical components show significant differences (Table 1). During E1, the average fractions of the rich-K, BB, OC, Ammonium, EC-aged, and Dust particle are 29.0%, 11.5%, 8.1%, 17.5%, 10.0% and 20.3%, respectively, different from



39.3%, 14.2%, 10.0%, 13.5%, 17.2%, and 1.3% respectively, during E2. It can be seen that
the Dust particle is the major changed factor, which is 93.6% lower during E2 than E1,
whereas the EC-aged particle shows a reversible of 72.0% higher during E2. The rich-K, BB,
and, OC particles show 22.9%-35.5% differences between the two episode periods. For the
air mass clusters (Fig. S8), E1 and E2 exhibit minor differences, mostly originating from
northern Myanmar and the Sino-Burmese border, but not identical regions. The Dust particles
that are much lower during E2 than E1 could be explained by higher WS (on average of 2.7 ±
1.0 m/s versus 0.4 ± 0.5 m/s) (Fig. S10) and PBL height (771 ± 717 m versus 560 ± 549 m).
The Dust particles are mainly formed by re-suspension in the local areas. In addition, the
quick thrown-up dust belongs to more coarse size particles, which are out of the detection
range of the SPAMS. However, due to the larger dust particles deposited more easily under
the low WS and the stagnant air conditions during E1, more suspended dust particles of small
size fall in the detection range of SPAMS. Moreover, the increased PBL height and WS could
speed up the transportation of pollutants from multiple sources (e.g., traffic and biomass
burning emissions) to the observation site, leading to evaluate the EC-aged, rich-K, BB, and
OC particles. The decreased Ammonium particle during E2 is potentially explained by the
reductions in the secondary pollutant formation with declines of RH (from 73.9% ± 23.9% to
53.1% ± 14.9%), the oxidation capacities [$O_3$: from 82.3 ± 5.5 ppbv to 76.8 ± 8.4 ppbv; and
$NO_x$: from, 3.9 ± 0.8 to 2.7 ± 0.8 ppb), in comparison to those during E1.

In terms of particle size distribution, the peak value of the rich-K, BB, and EC-aged

show minor differences (< 80 nm) between two episode periods (Fig. 5a). Expressed in
relative proportion (Fig. 5b), the rich-K and BB particles exhibit bimodal distributions, while
a peak at < 300 nm affected by the primary emissions and > 300 nm associated with the aging
process (Li et al., 2022b; Bi et al., 2011). In addition, compared with that during E1, rich-K
particles distribute in a wider size range and remind in a high percentage (> 20%) at ~1 μm
during E2 due to the atmospheric aging of the airmass. A similar aging process leads to the
particle size growth of BB, OC, Ammonium, and EC-aged particles, as well as a wider size
distribution. Due to relatively low concentration, the size distribution of Dust particles greatly
fluctuates.

During E1, more than 50% of $^{97}HSO_4^-$ fractions are mixed in the rich-K (81%), OC





(62%), Ammonium (100%), EC-aged (98%) particles (Fig. 6), lower than in BB (37%) and
Dust (4%) particles. Dissimilar with E1, the number fraction of $^{97}HSO_4^-$ increases to 34%
during E2, potentially associated with the enhancement by secondary formation. However,
the mixing state of $^{195}H(HSO_4)_2^-$), $^{62}NO_3^-$, $NH_4^+$and oxalate fractions are similar between the
two episodes events. The fractions of DEA are significantly higher in E2 than E1 for
Ammonium (67% versus 31%) and EC-aged particles (48% versus 17%), mainly due to the
high hygroscopic behavior (i.e., higher RH) (Sorooshian et al., 2007).

Photochemical oxidation and aqueous-phase reaction are the key formation pathways of

secondary species (Robinson et al., 2007; Link et al., 2017; Xue et al., 2014; Jiang et al.,
2019). Generally, the $O_x$ ($O_3 + NO_2$) concentration and RH serve as indicators of the degree of
photochemical oxidation (Wood et al., 2010) and aqueous-phase reaction (Ervens et al., 2011).
In this study, we chose the relative number fractions of $^{43}C_2H_3O^+$, $^{89}HC_2O_4^-$, $^{62}NO_3^-$, $^{97}HSO_4^-$,
and $^{18}NH_4^+$-containing particles to the total detected particles to indicate the secondary
formation (Liang et al., 2022), respectively. The correlations between the number fraction of
each secondary species with the oxidant concentrations ($O_x = O_3 + NO_2$) and RH are used to
reflect the formation pathways during the two events (Chen et al., 2016).

As shown in Fig. 7, compared with E1 and E2, the linear relationships between the

secondary aerosols and $O_x$ are the opposite. During E1, $^{43}C_2H_3O^+$, $^{89}HC_2O_4^-$, $^{97}HSO_4^-$, $^{18}NH_4^+$
show significant negative linear correlation with $O_x$ ($p < 0.01$), and the correlation strengths
range from moderate to strong (r = −0.51 ~ −0.81), except that $^{62}NO_3^-$ fraction shows a
certain higher but has no significant correlation with $O_x$ (r = 0.33, $p > 0.05$). This might be
influenced by the pollutant dispersion with the increased PBL height when $O_x$ was evaluated
(Fig. S9), offsetting the relatively low secondary formations (i.e., both precursors and local
anthropogenic emissions are low) near the study location (Li et al., 2016). However, the
number fraction of $NO_3^-$ exhibits an upward trend with the increase of $O_x$ concentration, due
to the rise of $NO_2$ concentration (Fig. S9). During E1, the increase of $NO_3^-$ could be generally
ascribed to the local $NO_2$ emission, while the declines of other secondary components might
be due to the reduced contribution of regional transportation to the other precursors. During
E2, $^{43}C_2H_3O^+$ has less significant correlation with $O_x$ (r = 0.37, $p > 0.05$), but with strong
correlations with $^{89}HC_2O_4^-$, $^{97}HSO_4^-$, and $^{18}NH_4^+$ (r = 0.81~0.92, $p < 0.01$). It should be noted





that $^{62}NO_3^-$ has a strong negative correlation (r =–0.85, $p$ < 0.01) with $O_x$. These results
suggest that photo-oxidation reactions have promoted secondary formation, among which the
rate of $HSO_4^-$ formation (slop = 0.017) is the highest. Increased with $O_x$ concentration, the
secondary organic species of $C_2H_3O^+$ (18%-28%) imperceptibly raise, and the oxalate also
increases by 7%-20%. This result shows that the secondary organic species were different in
the capacity of atmospheric oxidation formation, the obviously formed and accumulated
secondary acids might be due to the formation of ammonium oxalate by adsorption of $NH_3$
gas (Sullivan et al., 2007; Nie et al., 2012; Kawamura and Bikkina, 2016; Lin et al., 2019).
Increasing with $O_x$ concentration, the relative fraction of $NO_3^-$ decreases, attributed to its
relatively volatile and difficult remote transport during the aging process, and the formation
of organic nitrate. The previous study proves that the formations of organic nitrate species
(such as $^{27}CHN^+$, $^{30}NO^+$, $^{43}CHO_1N^+$, and $CHO_xN^+$) through the $NO+RO_2$ pathway dominate
80% of the total nitrate in tropical forested regions during summertime (Alexander et al.,
2009). Aruffo et al (2022) also found that low $NO_x$ (e.g., < 6 ppb) (2.3±0.8 ppbv in this study)
could even promote the particle-phase partitioning of the lower volatility of organonitrates.
Fig. 8 illustrates that $^{43}C_2H_3O^+$, $^{89}HC_2O_4^-$, $^{97}HSO_4^-$, and $^{18}NH_4^+$ have moderate to strong
positive correlations with RH (r = 0.70~0.81, $p$ < 0.01 or 0.05) during the two episode events,
except that $^{43}C_2H_3O^+$ during E2 ($p$ = 0.48) and $^{89}HC_2O_4^-$ during E1 ($p$ = 0.12). Furthermore,
$^{62}NO_3^-$ fraction also has no obvious changes and insignificant correlation with RH during E1
($p$ = 0.43) and presents a moderate negative correlation with RH (r = 0.69, $p$ < 0.01) during
E2. As shown in Fig. 8e, the aqueous formation rate of $HSO_4^-$ is higher during the E1 (slop is
0.014) than in E2 (slop is 0.009) due to the low volatile and high hygroscopicity of sulfate
(Wang et al., 2016; Zhang et al., 2019b; Sun et al., 2013). More favorable meteorological
conditions during E1, including lower WS (0.08 ± 0.08 m s$^{-1}$) and temperature (3.9 ± 0.8°C),
and higher RH (93.4 ± 7.6%), lead to the higher formation rate of $HSO_4^-$ than that during E2
(with meteorological parameters of 2.4 ± 0.8 m s$^{-1}$, 6.9 ± 1.2°C, 60.7 ± 8.7%, respectively).
The $NH_4^+$ species also have a greater production rate during the E1 (slop is 0.005) and E2
(slop is 0.006) due to the high RH. More abundant $NH_3$ precursors transported from the
surrounding under the high WS and acidic anion fraction (i.e., sulfate, and oxalate) advance
the $NH_4^+$ formation. Compared with those during E1, the secondary organic species (e.g.,





$C_2H_3O^+$ and $HC_2O_4^-$) show inversed generation rates during E2. In addition, $C_2H_3O^+$ shows a
strong correlation with RH (r = 0.70, $p < 0.05$) during E1 (slop is 0.003) but has insignificant
correlation during E2. However, the $HC_2O_4^-$ fraction has a slight higher (9.7%-13.1%) during
E1 and increases of correlation with RH (r = 0.81, $p < 0.01$) during E2 (slop is 0.003). These
could be explained by the elevated ammonium oxalate and its precursor ((i.e., $^{59}C_2H_3O_2^-$,
$^{71}C_3H_3O_2^-$, $^{73}C_2HO_3^-$) concentrations from biomass burning (Ervens et al., 2011; Li et al.,
2022b). The linearity between $^{62}NO_3^-$ and RH (r = 0.69, $p < 0.01$) decreases during E2,
mostly due to the low $NO_2$ concentration (2.6 ± 0.7 ppb) which further decreases with
elevating $O_3$ (Fig. S10). Meanwhile, high RH could promote organonitrates formation (Fang
et al., 2021; Fry et al., 2014). No obvious change and insignificant correlation between $NO_3^-$
and RH during E1, potentially attributed to the decreases of $NO_2$ concentration (3.7 ± 0.4 ppb)
in the local atmosphere.
**4    Conclusions**

This study presents the chemical composition, size distribution, mixing state, and

secondary formation of individual particles in the southeastern margin of TP, China during
the pre-monsoon season using a high-resolution SPAMS. The finding shows that the rich-K
(30.9%) and BB types (18.7%) are the two dominant aerosol particles in the remote area;
followed by the OC (12.8%), Ammonium (11.9%), EC-aged (10.9%), and Dust (10.7%) types;
the NaK-SN, Metal and Others types contributed 0.3–2.8% to the total amibent particles. By
interpreting the mass spectra and diurnal trends, the major particle types are mainly from
traffic emission, biomass burning, secondary formation, and fly ash, while the dynamics of
the PBL height could also affect the contributions of these particles. The observed change in
the number fraction of the particle types was mainly influenced by air masses (97.61% of the
total trajectories) from northeastern Myanmar, and significantly contributed to rich-K and BB
types. The particle types show distinct size distributions. The two most critical particle types
of rich-K and BB appear in a unimodal pattern, the fractions of OC and EC-aged gradually
decrease with the increase of the particle sizes, but Ammonium and Dust types show the
opposite. Sulfate is the major secondary species and is highly mixed with rich-K, Ammonium,
and EC-aged types. Nitrate has a relatively low mixing ratio due to its higher volatility than
sulfate during regional transportation, except for BB and OC types. During the entire study





campaign, two air episodes with the high number concentration particle occurred but with
significant differences in each particle fraction due to the different meteorological conditions
(RH, WS, etc.). The results of the formation mechanism of secondary species demonstrate
that the formation capacity of atmospheric oxidation is affected by the PBL height, but the
relative humidity (RH) could promote the formation of secondary species, especially $^{97}HSO_4^-$
and $^{18}NH_4^+$. The results of this study provide useful information concerning the detailed
characteristic of aerosol components, size distribution, and mixing states in the southeast TP,
and highlight the importance of the cross-border transport and formation mechanism of
aerosols in high-altitude regions.





Data availability. The data presented in this study are available at the Zenodo data archive https://doi.org/10.5281/zenodo.7336857.

Competing interests. The authors declare that they have no conflict of interest.

Author contributions. QW and JC designed the campaign. WR conducted field measurements. LL, QW, JT, and YZ made data analysis and interpretation. LL and QW wrote the paper. All the authors reviewed and commented on the paper.

Acknowledgments. The authors are grateful to the staff from Lijiang Astronomical Station for their assistance with field sampling. The authors are also grateful to Weikang Ran, Yonggang Zhang, and other staff for the field observation.

Financial support. This work was supported by the National Natural Science Foundation of China (41877391), the Second Tibetan Plateau Scientific Expedition and Research Program (STEP) (2019QZKK0602), and the Youth Innovation Promotion Association of the Chinese Academy of Sciences (2019402).

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



Table 1. The number concentrations, average percentages and characteristic ions of nine types
of particles during the entire study period, and the average percentages of the major six
particle types during two episodes.

| Type | Number count | Fraction in total (%) | Episode 1 (%) | Episode 2 (%) | Tracer ions |
|---|---|---|---|---|---|
| rich-K | 151040 | 30.9 | 29.0 | 39.3 | $^{39}K^+$, $^{26}CN^-$, $^{42}CNO^-$, $^{46}NO_2^-$, $^{62}NO_3^-$, $^{97}HSO_4^-$ |
| BB | 91322 | 18.7 | 11.5 | 14.2 | $^{39}K^+$, levoglucosan ($^{45}CHO_2^-$, $^{59}C_2H_3O_2^-$, $^{71}C_3H_3O_2^-$, $^{73}C_3HO_3^-$), $^{26}CN^-$, $^{35,37}Cl^-$, $^{42}CNO^-$, $^{46}NO_2^-$, $^{62}NO_3^-$, $^{97}HSO_4^-$ |
| OC | 62446 | 12.8 | 8.1 | 10.0 | $^{27}C_2H_3^+$, $^{37}C_3H^+$, $^{38}C_3H_2^+$, $^{39}K^+/C_3H_3^+$, $^{43}C_2H_3O^+$, $^{51}C_4H_3^+$, $^{26}CN^-$, $^{42}CNO^-$, $^{46}NO_2^-$, $^{62}NO_3^-$, $^{97}HSO_4^-$ |
| Ammonium | 58317 | 11.9 | 17.5 | 13.5 | $^{12}C^+$, $^{18}NH_4^+$, $^{39}K^+$, $^{58}C_2H_5NHCH_2^+$, $^{97}HSO_4^-$, $^{195}H(HSO_4)_2^-$ |
| EC-aged | 53337 | 10.9 | 10.0 | 17.2 | $C_n^\pm$ (n =1 ~ 5), $^{39}K^+$, $^{97}HSO_4^-$ |
| Dust | 52533 | 10.7 | 20.3 | 1.3 | $^{40}Ca^+$, $^{56}CaO^+$, $^{16}O^-$, $^{17}OH^-$, $^{76}SiO_3^-$, $^{79}PO_3^-$ |
| NaK-SN | 13726 | 2.8 | na | na | $^{23}Na^+$, $^{39}K^+$, $^{62}NO_3^-$, $^{97}HSO_4^-$ |
| Metal | 4672 | 1.0 | na | na | $^{51}V^+$, $^{56}Fe^+$, $^{64,66,68}Zn^+$, $^{206,207,208}Pb^+$ |
| Others | 1580 | 0.3 | na | na | No obvious characteristic peaks |




**Figure captions:**

**Figure 1.** Box and whisker diurnal plots of the number concentration of the main particle types (a) rich-Potassium (K), (b) Biomass burning (BB), (c) Organic carbon (OC), (d) Ammonium, (e) Element carbon (EC)-aged, (f) Dust in hourly resolution. The lower, middle, and upper lines of the boxes denote the 25th, 50th, and 75th percentiles. The lower and upper whiskers show the 10[th] and 90[th] percentiles, respectively. Average values are shown in white dots.

**Figure 2.** Maps of the mean HYSPLIT back trajectory clusters (72 h) at the height of 500 m during the whole field observation; the table embedded in the figure is the number concentration and relative fraction of the main six particle types in each cluster.

**Figure 3.** Size distributions of (a) the total number particle counts, (b) the relative percentages (%) of the total particles for nine groups during the sampling campaign.

**Figure 4.** Number fractions of secondary markers associated with the six particle types (rich-K, BB, OC, Ammonium, EC-aged, Dust). Secondary species include sulfate ($^{97}HSO_4^-$), sulfuric acid ($^{195}H(HSO_4)_2^-$), nitrate ($^{62}NO_3^-$), ammonium ($^{18}NH_4^+$), DEA (diethylamine, $^{58}C_2H_5NHCH_2^+$), and oxalate ($^{89}HC_2O_4^-$) ions.

**Figure 5.** Size distributions of the (a, c) number concentrations and (b, d) fractions of the major six particle types (rich-K, BB, OC, Ammonium, EC-aged, Dust) during two episodes of (a,b ) E1 and (c, d) E2.

**Figure 6.** Number fractions of secondary markers associated with the six particle types (rich-K, BB, OC, Ammonium, EC-aged, Dust) in two episodes events of E1 and E2: sulfate ($^{97}HSO_4^-$), sulfuric acid ($^{195}H(HSO_4)_2^-$), nitrate ($^{62}NO_3^-$), ammonium ($^{18}NH_4^+$), DEA (diethylamine, $^{58}C_2H_5NHCH_2^+$), and oxalate ($^{89}HC_2O_4^-$).

**Figure 7.** Correlations between the relative number fractions of the secondary species (a) $^{43}C_2H_3O^+$, (b) $^{89}HC_2O_4^-$, (c) $^{18}NH_4^+$, (d) $^{62}NO_3^-$, (e) $^{97}HSO_4^-$ and O$_x$ concentration during E1 (blue square) and E2 (red dot).

**Figure 8.** Correlations between the relative number fractions of the secondary species (a)





$^{43}C_2H_3O^+$, (b) $^{89}HC_2O_4^-$, (c) $^{18}NH_4^+$, (d) $^{62}NO_3^-$, (e) $^{97}HSO_4^-$ and relative humidity (RH)

during E1 (cyan dot) and E2 (orange square).

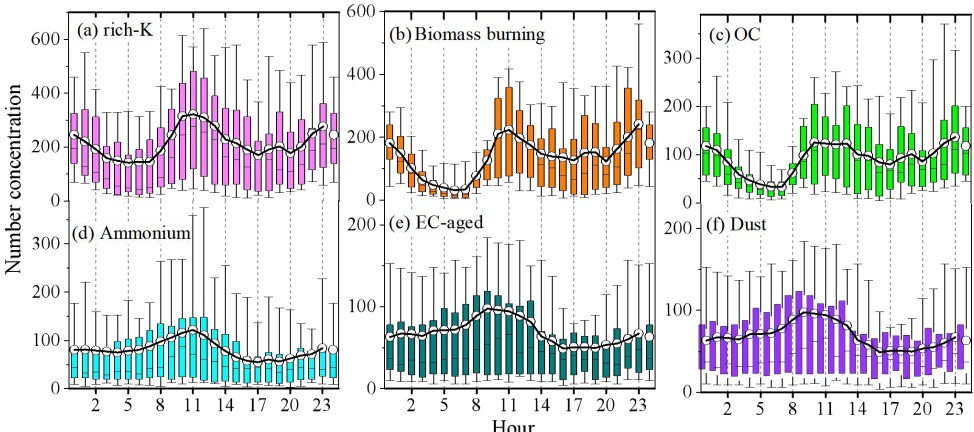

Figure 1. Box and whisker diurnal plots of the number concentration of the main particle types (a) rich-Potassium (K), (b) Biomass burning (BB), (c) Organic carbon (OC), (d) Ammonium, (e) Element carbon (EC)-aged, (f) Dust in hourly resolution. The lower, middle, and upper lines of the boxes denote the 25th, 50th, and 75th percentiles. The lower and upper whiskers show the 10th and 90th percentiles, respectively. Average values are shown in white dots.





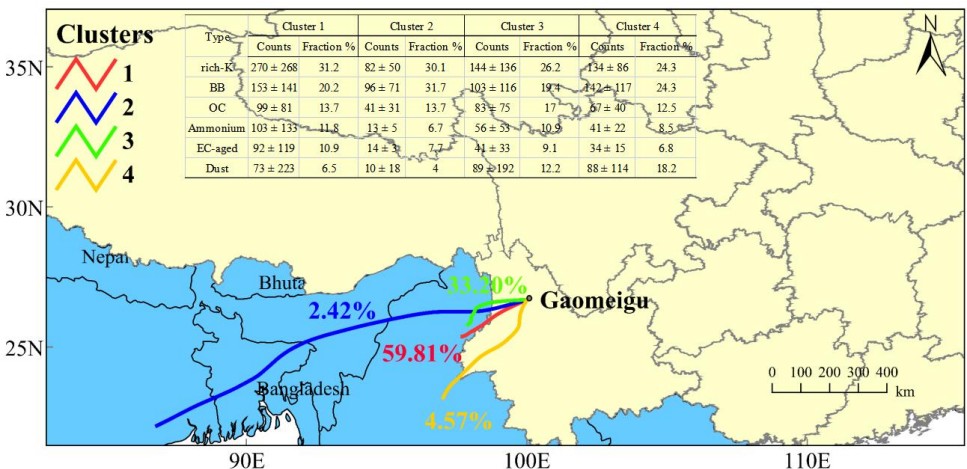


Figure 2. Maps of the mean HYSPLIT back trajectory clusters (72 h) at the height of 500 m during the
whole field observation; the table embedded in the figure is the number concentration and relative fraction
of the main six particle types in each cluster.



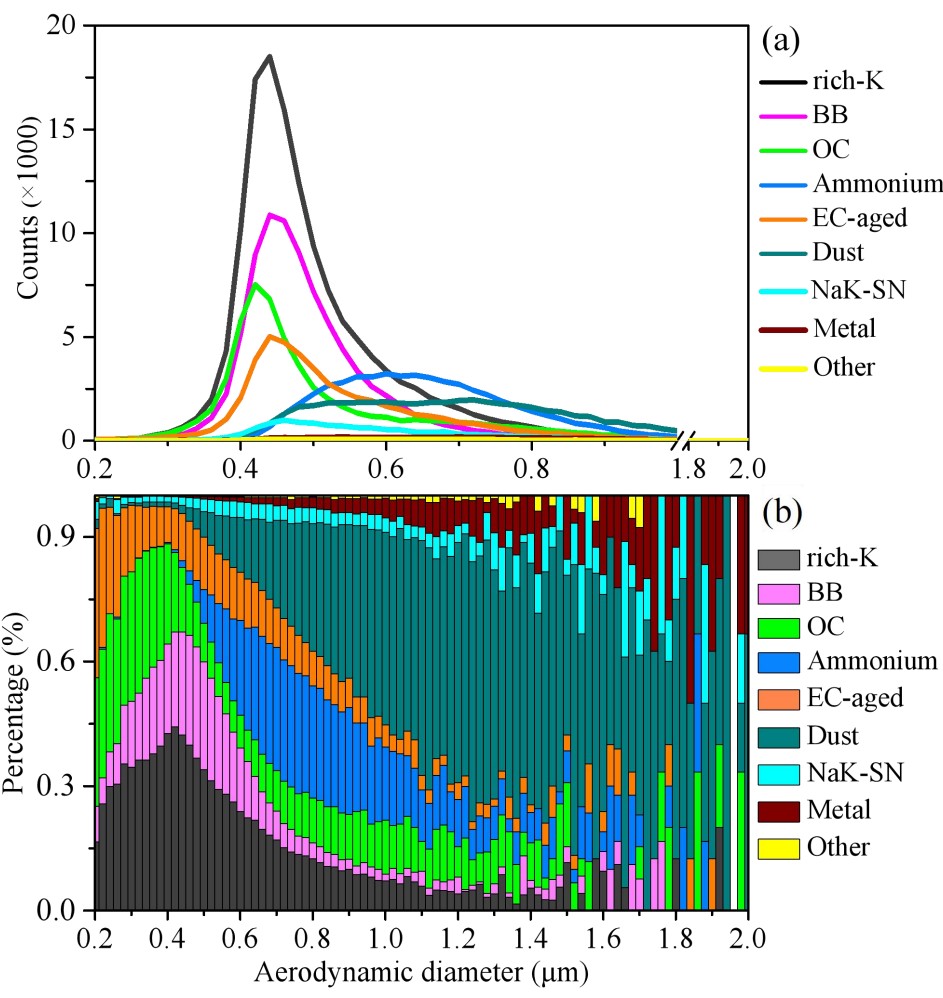

Figure 3. Size distributions of (a) the total number particle counts, (b) the relative percentages (%) of the
total particles for nine groups during the sampling campaign.




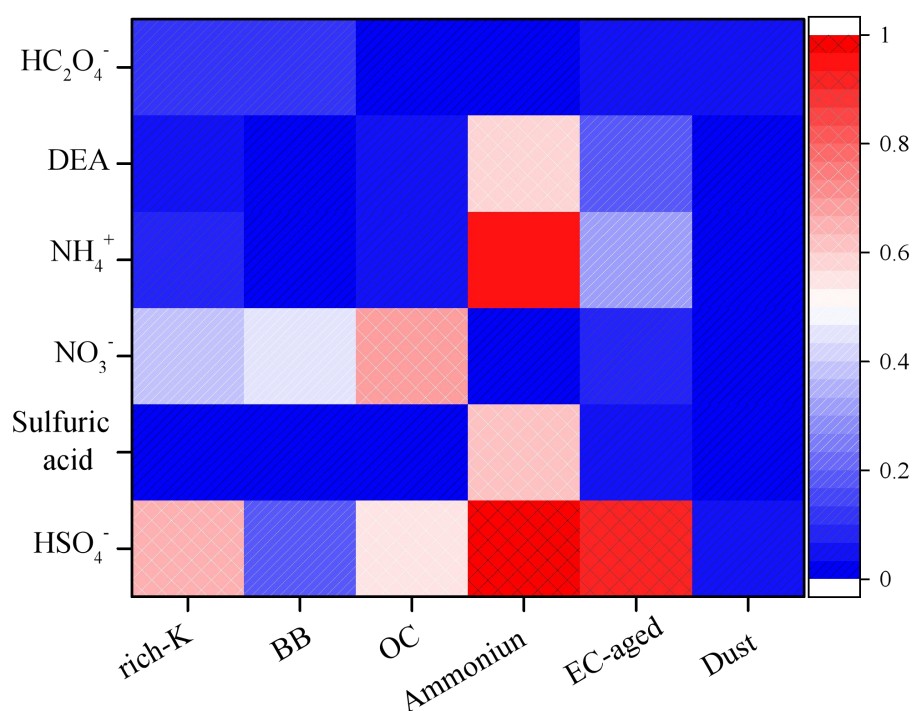

Figure 4. Number fractions of secondary markers associated with the six particle types (rich-K, BB, OC,
Ammonium, EC-aged, Dust). Secondary species include sulfate ($^{97}HSO_4^-$), sulfuric acid ($^{195}H(HSO_4)_2^-$),
nitrate ($^{62}NO_3^-$), ammonium ($^{18}NH_4^+$), DEA (diethylamine, $^{58}C_2H_5NHCH_2^+$), and oxalate ($^{89}HC_2O_4^-$) ions.



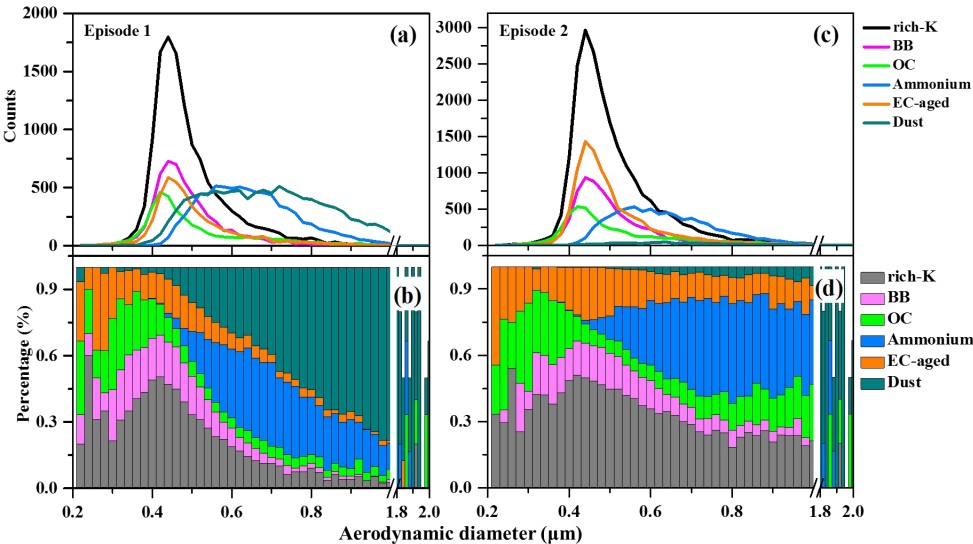


Figure 5. Size distributions of the (a, c) number concentrations and (b, d) fractions of the major six particle
types (rich-K, BB, OC, Ammonium, EC-aged, Dust) during two episodes of (a,b ) E1 and (c, d) E2.



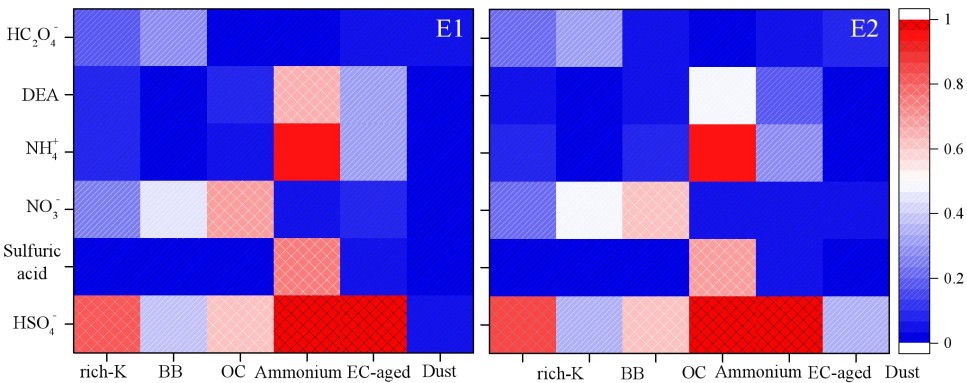

Figure 6. Number fractions of secondary markers associated with the six particle types (rich-K, BB, OC, Ammonium, EC-aged, Dust) in two episodes events of E1 and E2: sulfate ($^{97}HSO_4^-$), sulfuric acid ($^{195}H(HSO_4)_2^-$), nitrate ($^{62}NO_3^-$), ammonium ($^{18}NH_4^+$), DEA (diethylamine, $^{58}C_2H_5NHCH_2^+$), and oxalate ($^{89}HC_2O_4^-$).

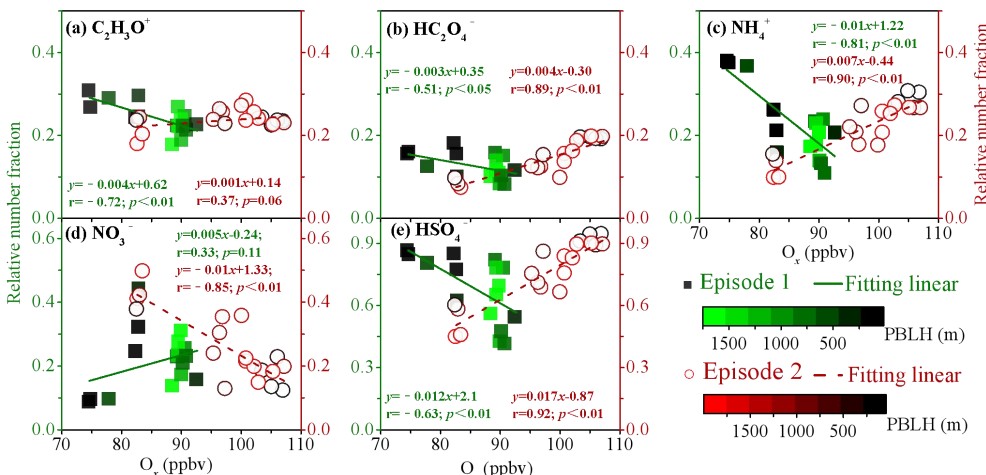


Figure 7. Correlations between the relative number fractions of the secondary species (a) [43]C$_2$H$_3$O$^+$, (b)
[89]HC$_2$O$_4^-$, (c) [18]NH$_4^+$, (d) [62]NO$_3^-$, (e) [97]HSO$_4^-$ and O$_x$ concentration during E1 (blue square) and E2 (red
dot).



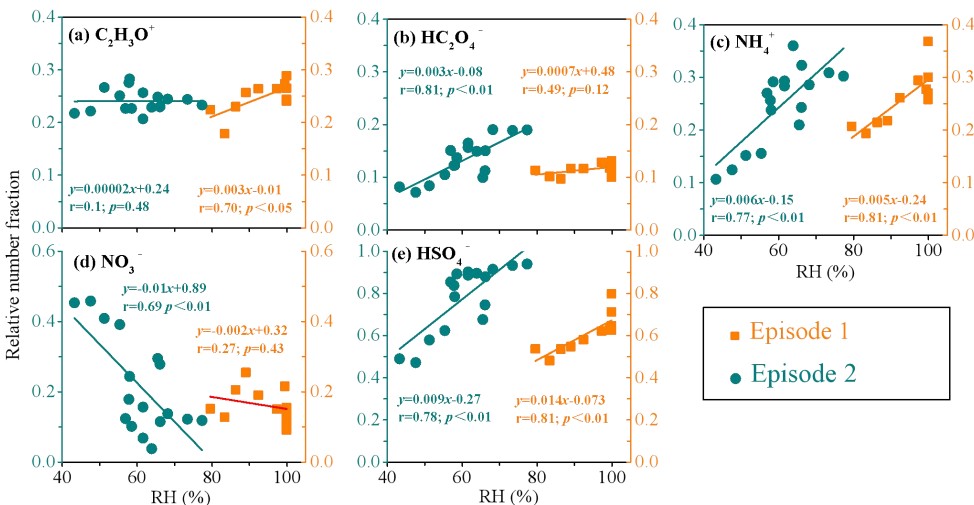

Figure 8. Correlations between the relative number fractions of the secondary species (a) [43]$C_2H_3O^+$, (b) [89]$HC_2O_4^-$, (c) [18]$NH_4^+$, (d) [62]$NO_3^-$, (e) [97]$HSO_4^-$ and relative humidity (RH) during E1 (cyan dot) and E2 (orange square).