# Peer review of "In-depth study of the formation processes of single atmospheric particles in"

_Atmospheric Chemistry and Physics, 2022_

## Author Comment (AC1)

**Responses to Referee #1:**

This paper studied the chemical compositions and mixing states of single particles in a rural site in the southeastern margin of the Tibetan Plateau (TP). The major particle types and size distributions of single particles were discussed, and the results of backward trajectories were coupled to investigate the regional impact on the formation of single particles in the sampling site. Two episodes were selected to discuss the transport and secondary formation processes of single particles. In addition, the linear regressions between several marker ions and RH, $O_x$ were explored to elucidate the formation processes of these secondary species. Generally, in view of the lack of field observation data in Tibetan Plateau, this study provides a good opportunity to investigate the mixing states and formation processes of single particles, which is of great significance to evaluate the influence of fine particles on the climate change in TP. However, several issues need to be addressed and some revisions are necessary before the acceptance of this manuscript.

**Response:** We highly appreciate the thoughtful and valuable suggestions by the reviewer, which are helpful for us to improve the quality of our manuscript. We have carefully addressed the comments in point-by-point form as shown below. Detailed responses to the comment are highlighted in blue, and the revised text is underlined in italics. Attached please also find the marked-up manuscript with tracked changes in the revised manuscript.

1. The size distributions of single particles from SPAMS should be scaled by other instruments such as SMPS, otherwise, the unscaled size distributions of single particles should be treated carefully, which mainly referred to the relative changes of same type particles at different period. The comparison of different type particles and quantitative description of size patterns are usually inaccurate. Authors presented many results of size distributions, but the length of discussions should be reduced and some expressions should be revised.

**Response:** Thank you for pointing out the unsuitable comparison. After considering the comments, we simplified the description of particle size distribution. As shown in Section 3.2 (new Lines 270-297), we also reduced the comparison and quantitative descriptions of the size distribution of different types of particles, due to their scales were not fully adjusted with either instruments or sampling periods. Those changes can be seen in the revised manuscript with "tracked change". Besides, due to the substantial changes that have been made, all the revisions have not been shown here, but the critical ones are listed as below.

Section 3.2 in Lines 270-297:
*"The aerodynamic size distributions of all particle types are shown in Fig. 3. According to the characteristics of the average MS (Text S1 and Fig. S3), rich-K, BB, OC and EC-aged particles originated from the similar sources of*

*vehicle emission or solid-fuel combustion. Their size distribution thus presents within a small-scale (~440 nm) (Fig. 3a). However, the relative percentage of each particle type is distinct with different size ranges, possibly due to the unique atmospheric processing. For example, as shown in Fig. 3b, the proportions of rich-K and BB types increase along with the increases in particle size from 200 to 420 nm but then decrease. OC and EC-aged types are mainly distributed in relatively small particle sizes, and their proportions gradually decrease when the size ranges become larger. Ammonium and Dust types are mainly distributed in large sizes of ~600 nm (Fig. 3a). The proportion of Ammonium particles gradually increases with the increase of particle size and peaks at 740 nm, the relatively large size distribution is ascribed to the intense atmospheric aging during regional transport (Text S1). The proportion of Dust particles gradually increases with a size > 560 nm and peaks at 1.48 μm. This is consistent with the fact that dust is a coarse particle, generally formed at the roadside and fly ash.*

*Compared with the total particle size distribution, the peak values of the six main particle types show minor differences (< 80 nm) during the two different episode periods (Fig. S10a). In Fig. S10b, a relatively high proportion of the rich-K and BB particles exhibit bimodal distributions, while peaks at < 300 nm are affected by the primary emissions and > 300 nm are associated with the aging process (Li et al., 2022b; Bi et al., 2011). Hence, the percentage of the six particle types distribute in wider size ranges during E2 than E1 due to the more intensive atmospheric aging. Relatively greater fluctuation for the large-size fractions (> 1.1 μm) could be explained by the low particle concentration (a number less than 20). It should be pointed out that further application of this method would require a co-located particle-sizing instrument to scale the size-resolved particle detection efficiency. Both particle composition and size-dependent are the predominant impacting factors on the particle detection efficiency of the SPAMS (Wenzel et al., 2003; Yang et al., 2017; Healy et al., 2013)."*

2. Line 32-33: why the volatilization of nitrate would lead to more abundant of sulfate?

**Response:** Sorry for the confusion. Our original statement aims to describe that the more volatilized nitrate ($^{62}NO_3^-$) leads to the low mixing of nitrate in the TP particles during the transport process. In contrast to nitrate, less volatilized sulfate mixes with the TP particles. The statement has been revised as follows.

Line 32-34
*"Compared with the abundant sulfate ($^{97}HSO_4^-$), the low nitrate ($^{62}NO_3^-$) internally mixed in TP particles is mainly due to the fact that nitrate is more volatilized during the transport process."*

3. Line 34-35: not all of these secondary species showed strong linear regressions with RH and Ox from the discussion of Section 3.3. Authors should give a more precise conclusion.

**Response:** Suggestion taken. A detailed description of the formation mechanism of secondary species is shown in Section 3.3. The linear regressions between each secondary species and $O_x$ are different during E1 and E2. The number fractions of these secondary species have moderate to strong positive correlations with RH during the two episodes, except that $^{43}C_2H_3O^+$ during E2 and $^{89}HC_2O_4^-$ during E1. Furthermore, $^{62}NO_3^-$ has an insignificant correlation with RH during E1 and presents a moderate negative correlation with RH during E2. Conclusively, the formation capacity of atmospheric oxidation is presumably weakened by the convective transmission and strengthened by the regional transport in TP, but RH could significantly promote the formation of secondary species, especially $^{97}HSO_4^-$ and $^{18}NH_4^+$. The sentence has been rewritten in the Abstract as follows.

> Line 34-38
> *"The formation mechanism of secondary speciation demonstrates that the formation capacity of atmospheric oxidation is presumably affected by the convective transmission and the regional transport in TP. However, the relative humidity (RH) could significantly promote the formation of secondary species, especially for $^{97}HSO_4^-$ and $^{18}NH_4^+$."*

4. Many field studies via SPAMS have been reported in recent years, thus, authors should add some new references especially those published after 2018.

**Response:** Thanks for pointing out these. We have replaced the old references with more updated ones. For example, in Lines 91-96, new references are added as follows.

> Line 91-96:
> *"At the same time, aerosol time-of-flight mass spectrometry (ATOFMS) (Dall'Osto et al., 2014) and single particle aerosol mass spectrometer (SPAMS) (Zhang et al., 2020) are popular for characterizing atmospheric individual particles. These devices can determine the chemical compositions and size distributions of the particles in detail, such as the dynamic processes of chemical aging, mixing state, and transport of the aerosols (Liang et al., 2022; Li et al., 2022b; Zhang et al., 2019b)."*

**References:**
Li L., Wang, Q. Y., Zhang, Y., Liu, S. X., Zhang, T., Wang, S., Tian, J., Chen, Y., Hang Ho, S. S., Han, Y., and Cao, J.J.: Impact of reduced anthropogenic emissions on chemical

characteristics of urban aerosol by individual particle analysis, Chemosphere, 303, 135013, https://doi.org/10.1016/j.chemosphere.2022.135013, 2022b.

Liang, Z. C., Zhou, L. Y., Cuevas, R. A., Li, X. Y., Cheng,C. L., Li, M., Tang, R. Z., Zhang, R. F., Lee Patrick K. H., Lai, Alvin C. K., and Chan, C.K.: Sulfate Formation in Incense Burning Particles: A Single-Particle Mass Spectrometric Study, Environ. Sci. Technol. Lett., 9, 718–725, https://doi.org/10.1021/acs.estlett.2c00492, 2022.

Zhang, G. H., Lin, Q. H., Peng, L., Yang, Y. X., Jiang, F., Liu, F. X., Song, W., Chen, D. H., Cai, Z., Bi, X. H., Miller, M., Tang, M. J., Huang, W. L., Wang, X. M., Peng, P. A., Shen, G. Y.: Oxalate Formation Enhanced by Fe-Containing Particles and Environmental Implications, Environ. Sci. Technol., 53, 1269–1277, https://doi.org/10.1021/acs.est.8b05280, 2019b.

Zhang, G. H., Lian, X. F., Fu, Y. Z., Lin, Q. H., Li, L., Song, W., Wang, Z. Y., Tang, M. J., Chen, D. H., Bi, X. H., Wang, X. M., and Sheng, G. Y.: High secondary formation of nitrogen-containing organics (NOCs) and its possible link to oxidized organics and ammonium, Atmos. Chem. Phys., 20, 1469–1481, https://doi.org/10.5194/acp-20-1469-2020, 2020.

5. There are many grammatical errors and unprofessional descriptions in the manuscript, such as: line 47-48, "making their impact on the air more uncertain"; line74-75, "Atmospheric aerosols also can influence the properties and life span of clouds as cloud condensation nuclei"; line 81, "Most studies have focused on the influence of optical properties"; line 92, "with a high temporal resolution"; line 95, "AMS/ACSM mainly used to provide"; line 102, remove "full" from "determine the full chemical composition"; line 107, "The shortage of information"; line 115, "pre-monsoon, to continuously (i) investigate"; line 123, "2.1 Observation site"; line 129, "The villagers make a living by farming (e.g., potato and autumn rape), and biomass is the main residential fuel"; line 143, "a detection moment"; line 194, "Differently, few". In general, the related grammatical errors are not limited to these examples, authors should carefully revise the manuscript to meet the quality of ACP.

**Response:** Thanks for pointing out the mistakes. The revised manuscript has been edited and proofread by a native English speaker. All related grammatical errors shown in this comment have been corrected as follows.

Line 47-49:
*"After further coating through coagulation, condensation and photochemical oxidation, its sizes, chemical compositions, mixing states, and optical properties would change greatly, leading to its influence in the atmosphere more uncertain"*

Line 68-70:
*"Atmospheric aerosols also can act as cloud condensation nuclei to impact the local hydrological cycles and monsoon patterns by changing the microphysical properties and life span of clouds."*

Line 73-75:

*"Most studies have focused on the optical characteristics within the TP; however, only a few research has been conducted on aerosol components."*

Line 85:

*"More advanced aerosol online measurement equipment with high-time resolution...."*

Line 85-91:

*"More advanced aerosol online measurement equipment with high-time resolution, such as the aerosol chemical speciation monitor (ACSM) and aerosol mass spectrometer (AMS) (Ng et al., 2011; Canagaratna et al., 2007) are mainly used to achieve online observation datasets of non-refractory submicron aerosol (including the mass concentration of sulfate, nitrate, ammonium, chloride, and organic; and their corresponding mass spectral). This is beneficial to recognize the dynamic processes of source emission in the atmosphere (Du et al., 2015; Zhang et al., 2019a)."*

Line 93-96:

*"These devices can determine the chemical composition and size distribution of the particles in detail, such as the dynamic processes of chemical aging, mixing state, and transport of the aerosols (Liang et al., 2022; Li et al., 2022b; Zhang et al., 2019b)."*

Line 96-100:

*"To the best knowledge, the advanced measurement device has not yet been applied for the studies conducted in TP, leading to a lack of in-depth research on the $PM_{2.5}$ pollution in TP, especially in the southeastern margin, which hinders our understanding of the distribution characteristics and formation mechanism of aerosol components in high-altitude regions."*

Line 104-106:

*"In this study, continuous field observation of individual particles (SPAMS) was deployed on the southeastern margin of the TP during the pre-monsoon period, to (i) investigate..."*

Line 113:

*"2.1 Sampling site"*

Line 119-120:

*"Villagers earn a living by farming (e.g., potato and autumn rape), and biomass is the major domestic fuel (Li et al., 2016)."*

Line 133-135:

*"In summary, a velocity, a detection time, and an ion mass spectrum are recorded for each ionized particle, while there is no mass spectrum for not ionized particles."*

Line 187-190:
*"The difference is that few researchers can capture the high proportion of Ammonium particles as shown in this study (Shen et al., 2017; Xu et al., 2018), which is ascribed to the conversion of ammonia (NH₃) precursor emitted from large-scale agricultural activities and mountain forest (Engling et al., 2011; Li et al., 2013)."*

6. Line 49-59: these sentences in the introduction were repetitive and should be reduced in length.

**Response:** We agreed with this point. The statements have been revised as follows.

Line 47-54:
*"After further coating through coagulation, condensation and photochemical oxidation, its sizes, chemical compositions, mixing states, and optical properties would change greatly, leading to its influence in the atmosphere more uncertain (Jacobson, 2002; Zaveri et al., 2010; Matsui, 2016; Budisulistiorini et al., 2017; Ma et al., 2012). Currently, the influences of the complex chemical components on aerosol size and mixing state show large regional differences due to the variations in the pollution sources, atmospheric formation mechanism and meteorological conditions, which have been widely studied in an urban area at a low altitude (Pratt et al., 2011; Liu et al., 2020a; Xu et al., 2017; Wang et al., 2022)."*

**Reference:**
Xu, J., Li, M., Shi, G., Wang, H., Ma, X., Wu, J., Shi, X., and Feng, Y.: Mass spectra features of biomass burning boiler and coal burning boiler emitted particles by single particle aerosol mass spectrometer, Sci. Total Environ., 598, 341–352, https://doi.org/10.1016/j.scitotenv.2017.04.132, 2017.

7. Line 94-95: The AMS and SPAMS both have its advantages to conduct the researches in aerosols, so you can directly present their application in the aerosol study instead of pointing out the things they can't do.

**Response:** The description of the AMS/ACSM deficiencies has been deleted. Their application in the aerosol study has been added to the revised manuscript. We have rewritten this text as follows.

Line 85-91:

*"More advanced aerosol online measurement equipment with high-time resolution, such as the aerosol chemical speciation monitor (ACSM) and aerosol mass spectrometer (AMS) (Ng et al., 2011; Canagaratna et al., 2007) are mainly used to achieve online observation datasets of non-refractory submicron aerosol (including the mass concentration of sulfate, nitrate, ammonium, chloride, and organic; and their corresponding mass spectra). This is beneficial to recognize the dynamic processes of source emission in the atmosphere (Du et al., 2015; Zhang et al., 2019a)."*

8. Line 165-168: the "Aging Element Carbon (EC-aged)" should be "aged elemental carbon (EC-aged)". "Potassium-containing (NaK-SN)," doesn't match its abbreviation, and what is the difference of this type with "Potassium-rich (rich-K)"?

**Response:** We do apologize for the unclear description of the "NaK-SN" particles in the original manuscript. Particles containing the strongest $K^+$ (m/z $^{39}K^+$) signal accompanies by a significant sulfate signal (m/z $^{97}HSO_4^-$) in positive MS, and nitrate signals (m/z $^{46}NO_2^-$, $^{62}NO_3^-$) in the negative MS, are identified as Potassium-rich (rich-K) (Fig. S3a). Compared with the rich-K particles, the NaK-SN particles also contain abundant sodium ion signal in the positive MS, but more abundant with sulfate shown in the negative MS. In addition, the intensities of nitrate fragments are significantly lower. Considering such differences between rich-K and NaK-SN particles, they might come from unique origins (e.g., sources and distribution characteristics). Therefore, the two particle types are resolved separately. The statements have been revised as below.

Line 157-161:
*"Finally, eight major particle clusters [i.e., potassium-rich (rich-K), biomass burning (BB), organic carbon (OC), Ammonium, aged element carbon (EC-aged), Dust, sodium (Na)-potassium (K)-containing (NaK-SN), and iron (Fe)-lead (Pb)-containing (Metal)] with distinct chemical patterns were manually combined, which represent ∼99.7 % of the population of the detected particles."*

9. Line 177: For the trajectory clusters analysis, I don't think the height of 500 m is reasonable to elucidate the transportation of air masses in consideration of the mountains and plateau surrounding the sampling site.

**Response:** HYSPLIT converts the vertical layers from the original coordinate system into its terrain-following coordinate system (sigma) and directly uses the data contained in meteorological files to the calculated trajectory (Draxler and Hess, 1998). GDAS data is derived from the sigma coordinate as well. The surface in the terrain-following coordinate system is consistent with the coordinate surface, so solves the problem of modeling near mountains area (Phillips, 1965). The trajectory position could be slightly different because of using different meteorological files

which origin from the data discrepancy in the different meteorological files, but this method has been used over complex terrain with various meteorological data (Khan et al., 2010; Burley and Bytnerowicz 2011; Qu et al 2015; Wang et al., 2015, 2019).

To further determine if the trajectory would be impacted by the surface rising. We have performed sensitivity tests for the starting height. Fig. R1 show the 72-h backward trajectories based on the assumed starting height of 800 m and 1000 m, respectively. Although there is some difference in the 72-h backward trajectories between the three different heights (i.e., 500, 800 and 1000 m), the main transport pathways are similar. We finally ran the trajectory at 500 m is considerately representative of the average planetary boundary level (~590 m). This has been also adopted in many other studies (Zhang et al., 2019; Lu et al., 2012; Liu et al., 2021; Tian et al., 2023).

Four groups of air masses are identified based on their transport pathways (Fig. R1). The maps show that the major trajectory clusters are similar at different arrival heights, representing that the height of 500 m could effectively capture the large-scale flow patterns.

[Figure]

[Figure]

Figure R1. Maps of mean trajectory clusters at arrival heights of 500, 800, and 1000 m above ground level during the campaign.


**Response:** The particle mass spectra for coal combustion, biomass burning, and vehicle emissions have strong signals for $^{97}HSO_4^-$. The digitized spectra for coal combustion (Fig. R2a) and biomass burning (Fig. R2b) are obtained from our in-lab experiments. The mass spectrum for vehicular emissions (Fig. Rxxc) was collected by Yang et al (2017). As shown in Fig. R2a, the mass spectra for the coal combustion show strong sulfate ($^{97}HSO_4^-$) signal, but the nitrate ($^{62}NO_3^-$) signal is almost absent. As shown in Fig. R2b and c, the mass spectra for biomass burning and vehicular emission contain higher $^{97}HSO_4^-$ than $^{62}NO_3^-$ signals. The comparison between the mass spectra of the sources and ambient particles sample indicates that the EC-containing particles mainly originated from motor vehicles and coal combustion in our study.

[Figure]

[Figure]

*Figure R2. Average digitized spectra for (a) coal combustion and (b) biomass burning emissions obtained with the SPAMS. Vehicle exhaust spectra (c) obtained by Yang et al (2017).*

After careful consideration and comparison, the related descriptions have been revised as follows.

Line 329-333:

*"Compared to the individual particle mixing state in urban or suburban areas that are located close to emission sources (Chen et al., 2016; Dall'Osto and Harrison, 2012; Zhang et al., 2017a; Li et al., 2022b), the high fractions of sulfate and ammonium at the high altitude area demonstrate a high degree of aging of the individual particles, whereas the low fraction of nitrate with high volatility indicates its loss during transportation processing."*

*"The number fractions of six markers in the four clusters were used to further investigate the impacts of regional transport. As shown in Fig. 5a and c, the dominant mixing ion types in each particle (except for Dust type) are similar among the four Clusters. For Cluster 1, the number fractions of $^{97}HSO_4^-$ and $^{89}HC_2O_4^-$ have larger values in five particle types (except for Dust type) than those*

*in other clusters. Similar to Cluster 1, 3 and 4 are impacted by regional transport from northeastern Myanmar. Moreover, the fractions of the six markers are similar in OC, Ammonium, and EC-aged types. However, $^{97}HSO_4^-$ is decreased in rich-K, BB, and Dust types, while $^{62}NO_3^-$ is incresed in rich-K and decreased in Dust types. As discussed in Section. 3.1, these phenomena demonstrate that the aging degree of Cluster 3 and 4 might be lower than that of Cluster 1. For Cluster 2, the fraction of $^{97}HSO_4^-$ is decreased in rich-K, BB, and EC-aged types but slightly increased in Dust type (Fig. 5f). Such pattern inverse the observations in rich-K, OC, and Dust types for $^{62}NO_3^-$ ions. These variations in Cluster 2 are more likely due to influences of biomass-burning activities from the surrounding area of the sampling site, rather than regional transport. Furthermore, Cluster 2 is associated with regional transport from northeastern India along the afternoon to nighttime (from 15:00 LT on May 11$^{th}$ to 07:00 LT on 12 May) which is favorable to the nitrate formation $N_2O_5$ by heterogeneous hydrolysis (Wang et al., 2017; Ding et al., 2021). However, these cases are infrequent, as only 2% of trajectories are associated with Cluster 2."*

[Figure]

Figure 5. Number fractions of secondary markers associated with the six particle types (i.e., rich-K, BB, OC, Ammonium, EC-aged, and Dust) in four clusters. Secondary markers include sulfate ($^{97}HSO_4^-$), sulfuric acid ($^{195}H(HSO_4)_2^-$), nitrate ($^{62}NO_3^-$), ammonium ($^{18}NH_4^+$), amine ($^{58}C_2H_5NHCH_2^+$), and oxalate ($^{89}HC_2O_4^-$).

[revised manuscript text omitted]

---

## Author Comment (AC2)

**Responses to Referee #2:**

In this work, Li et al. present single-particle aerosol mass spectrometry (SPAMS) measurements on the Tibetan Plateau. Measurements were taken over a month as part of an intensive campaign. In addition to SPAMS measurements, meteorological conditions were measured, as well as ozone and NOx. Almost 500,000 bi-polar, single-particle mass spectra were obtained. Spectra were subject to a clustering analysis. Air mass back trajectories were also calculated, and also were clustered. Two episodes of high particle concentrations are highlighted. From a cluster analysis of back trajectories, these episodes are from similarly sourced air, but they contain different fractions of particle types. The main difference is that there was more dust in Episode 1. Finally, the authors found relationships between the fraction of particles containing certain secondary aerosol markers and Ox and RH.This work is novel and is deserving of publication, but it requires major revisions in order to be published. See the major comments section for more details

**Response:** We sincerely appreciate all valuable comments and suggestions to our manuscript. We have carefully revised the relevant content. We do believe that the revised manuscript has been significantly improved after the revision. The point-to-point responses are shown, the modifications to the manuscript are marked in blue, and the revised text is underlined in italics. Attached please also find the marked-up manuscript with tracked changes.

**Major Comments**

1. There are many grammatical errors throughout this text. This reviewer strongly suggests that the authors carefully revise the manuscript to address these errors. Unfortunately, at this point, it makes it difficult to understand the author's interpretation of the results, and thus the paper suffers. Here are some examples from the abstract alone:

**Response:** All suggested typos and grammatical mistakes have been corrected. Meanwhile, the revised manuscript has been edited and proofread by a native English speaker. The critical revision has been shown in follows.

1. Line 19: "… which is a transport channel for pollutants from South Asia [to where?] during the pre-monsoon season."

Line 15-19:
*"The unique geographical location of the Tibetan Plateau (TP) plays an important role in regulating global climate change, but the impacts of the chemical components and atmospheric processing on the size distribution and mixing state of individual particles are rarely explored in the southeastern margin of the TP, which is a transport channel for pollutants from South Asia to the TP during the pre-monsoon season."*

2. Line 25: It is not clear what the "surroundings" means here, and the phrase "cross-border" here is either excessive or confusing.

Line 24-27:

*"Most particle types were mainly transported from the surroundings of the sampling site and Sino-Myanmar border; but the air masses from northeastern India and Myanmar show a greater impact on the number fraction of BB (31.7%) and Dust (18.2%) types, respectively."*

3. Line 27: I'm not sure what the "Besides" at the beginning of this sentence is referring to. I also think you should say either "episodes" or "events" here, but probably not "episodes events."

Line 28-31:

*"Then, the two episodes with high particle concentrations showed that the differences in the meteorological conditions in the same trajectory clusters could cause significant changes in chemical components, especially the Dust and EC-aged types, which changed by a sum of 93.6% and 72.0%, respectively."*

4. Line 29: The phrase "air clusters" is used here, but it has not been defined before, and it is not intuitive from the wording what it means.

Line 29:

We have exchanged the original clause "air clusters" for *"trajectory clusters"* in the revised version.

5. Line 30: I think you want to add a ", which" after "types" and a "," after "72%."

Line 30-31:

*"especially the Dust and EC-aged types, which changed by a sum of 93.6% and 72.0%, respectively."*

6. Line 34: I think "severed" is supposed to be "served."

Corrected.

2. Because there are many instances of grammatical errors throughout the text, the focus of the following comments will be of a general nature and will mainly be guided by the figures themselves; specific comments on the text in the main body of the paper will be withheld in anticipation of major revisions.

**Response:** The grammatical errors and unprofessional descriptions are thoroughly revised in this revised manuscript. The revised manuscript has been polished by a native English-speaking scientist.

3. The authors dedicate a lot of time in the results section interpreting the "size distributions." Unfortunately, it is difficult to interpret these results without some sense of the total aerosol size distribution (from an SMPS or OPC like

UHSAS, LAS, etc.). Your "size distribution," as presented, is sensitive to the detection efficiency of particles as a function of size. Furthermore, it may also be sensitive to particle shape. Both need to be accounted for prior to interpreting the size distribution.

**Response:** After consideration of the reviewer's comment, the descriptions of particle size distribution have been reduced. As shown in Section 3.2 (new Lines 270-297), the comparison and quantitative description of the size distribution of different types of particles have been deleted in the revised manuscript.

Line 270-297:

*"The aerodynamic size distributions of all particle types are shown in Fig. 3. According to the characteristics of the average MS (Text S1 and Fig. S3), rich-K, BB, OC and EC-aged particles originated from the similar sources of vehicle emission or solid-fuel combustion. Their size distribution thus presents within a small-scale (~440 nm) (Fig. 3a). However, the relative percentage of each particle type is distinct with different size ranges, possibly due to the unique atmospheric processing. For example, as shown in Fig. 3b, the proportions of rich-K and BB types increases along with the increases in particle size from 200 to 420 nm but then decrease. OC and EC-aged types are mainly distributed in relatively small particle sizes, and their proportions gradually decrease when the size ranges become larger. Ammonium and Dust types are mainly distributed in large sizes of ~600 nm (Fig. 3a). The proportion of Ammonium particles gradually increases with the increase of particle size and peaks at 740 nm, the relatively large size distribution is ascribed to the intense atmospheric aging during regional transport (Text S1). The proportion of Dust particles gradually increases with a size > 560 nm and peaks at 1.48 μm. This is consistent with the fact that dust is a coarse particle, generally formed at the roadside and fly ash.*

*Compared with the total particle size distribution, the peak values of the six main particle types show minor differences (< 80 nm) during the two different episode periods (Fig. S10a). In Fig. S10b, a relatively high proportion of the rich-K and BB particles exhibit bimodal distributions, while peaks at < 300 nm are affected by the primary emissions and > 300 nm are associated with the aging process (Li et al., 2022b; Bi et al., 2011). Hence, the percentage of the six particle types distribute in wider size ranges during E2 than E1 due to the more intensive atmospheric aging. Relatively greater fluctuation for the large-size fractions (> 1.1 μm) could be explained by the low particle concentration (a number less than 20). It should be pointed out that further application of this method would require a co-located particle-sizing instrument to scale the size-resolved particle detection efficiency. Both particle composition and size-dependent are the predominant impacting factors on the particle detection efficiency of the SPAMS (Wenzel et al., 2003; Yang et al., 2017; Healy et al., 2013)."*

In our study, the aerodynamic size range for SPAMS is 0.2-2.0 μm. A silicone dryer was also set up in front of the inlet system to avoid water vapor condensation which reduces the uncertainties in particle collection efficiency due to variable humidity (Matthew et al., 2008; Zhao et al., 2017). The above description has been supplemented in Section 2.2.

Line 139-141

*"A hollow silicone dryer was installed in front of the inlet. This reduces the uncertainty of particle collection efficiency due to the changes of humidity in sampled airs."*

According to the schematic diagram of the SPAMS (Fig. R3), the distance between two 532 nm lasers is known. In addition, the time of flight of the particle could be counted, while the particle velocities are thus calculated. Different sizes (i.e., 240, 320, 510, 740, 960 nm, 1.4 μm, and 2.0 μm) of standard polystyrene latex spheres (PSL) were used to calibrate the SPAMS. The calibration curve was developed as below.

$$y = C_1 + C_2 x + C_3 x^2 + C_4 x^3$$

*where y* is the particle aerodynamic size; *x* is the particle velocity. The particle size calibration parameters (i.e., $C_1$, $C_2$, $C_3$, and $C_4$) were obtained under the inlet pressure of 2.40 Torr. The continuous velocity of the particles was converted into the corresponding vacuum aerodynamic diameter.

According to the information shown by Li et al. (2011), the main influencing factor that could cause the size deviation of the measured particles is the inlet pressure. The inlet pressure must be controlled within the range of ± 0.05 Torr as small as possible. During the sampling period, if the critical orifice is contaminated, the inlet pressure could be altered. To avoid this issue, we have regularly used the supersonic cleaner to remove the contamination in the critical orifice by soaking it in an ethanol solution. In addition, we have controlled the inlet pressure between 2.35 and 2.45 Torr to ensure the accuracy of the data obtained by SPAMS during the entire observation period.

[Figure]

Figure R3. Schematic diagram of the SPAMS.

A publication reported that the particle-detection efficiency of the SPAMS exhibits a strong dependence on particle sizes and counts of different particle types in each size range, not absolute atmospheric concentrations (Yang et al., 2017). Although the SPAMS determines the number size distribution in a size range of 200-2000 nm, the detection efficiency is low at both ends of the size range. Therefore, quantifying the absolute contribution of each particle type to ambient particle number and mass concentration is difficult. Composition-dependent ionization efficiency and size-dependent particle detection efficiency are the predominant confounding factors (Allen et al., 2000; Reilly et al., 2000; Wenzel et al., 2003; Healy et al., 2013). The characteristics of aerodynamic size measured by SPAMS are statistical results, while the relative number fraction of different particle types in each size bin is significant. To clarify this concern, more descriptions of the size distribution using SPAMS have been added as follows.

Line 293-297:
*"It should be pointed out that further application of this method would require a co-located particle-sizing instrument to scale the size-resolved particle detection efficiency. Both particle composition and size-dependent are the predominant impacting factors on the particle detection efficiency of the SPAMS (Wenzel et al., 2003; Yang et al., 2017; Healy et al., 2013)."*

**Reference:**
Allen, J. O., Fergenson, D. P., Gard, E. E., Hughes, L. S., Morrical, B. D., Kleeman, M. J., Gross, D. S., Galli, M. E., Prather, K. A., and Cass, G. R.: Particle Detection Efficiencies of Aerosol Time of Flight Mass Spectrometers under Ambient Sampling Conditions, Environ. Sci. Technol., 34, 211–217, http://dx.doi.org/10.1021/es9904179, 2000.

Healy, R. M., Sciare, J., Poulain, L., Crippa, M., Wiedensohler, A., Prévôt, A.S.H., Baltensperger, U., Sarda-Estève, R., McGuire, M. L., Jeong, C. H., McGillicuddy, E., O'Connor, I. P., Sodeau, J. R., Evans, G. J., and Wenger, J. C.: Quantitative determination of carbonaceous particle mixing state in Paris using single-particle mass spectrometer and aerosol mass spectrometer measurements, Atmos. Chem. Phys., 13, 9479–9496, http://dx.doi.org/10.5194/acp-13-9479-2013, 2013.

Kane, D. B. and Johnston, M. V.: Size and Composition Biases on the Detection of Individual Ultrafine Particles by Aerosol Mass Spectrometry, Environ. Sci. Technol., 34, 4887–4893, https://doi.org/10.1021/es001323y, 2000.

Matthew, B. M., Middlebrook, A. M., and Onasch, T. B.: Collection Efficiencies in an Aerodyne Aerosol Mass Spectrometer as a Function of Particle Phase for Laboratory Generated Aerosols, Aerosol Sci. Technol., 42, 884–898, https://doi.org/10.1080/02786820802356797, 2008.

Wenzel, R. J., Liu, D.-Y., Edgerton, E. S., and Prather, K. A.: Aerosol time-of-flight mass spectrometry during the Atlanta Supersite Experiment: 2. Scaling procedures, J. Geophys. Res., 108, 8427, http://dx.doi.org/10.1029/2001jd001563, 2003.

Yang, J., Ma, S. X., Gao, B., Li, X. Y., Zhang, Y. J., Cai, J., Li, M., Yao, L. A., Huang, B., and Zheng, M.: Single particle mass spectral signatures from vehicle exhaust particles and the

source apportionment of on-line PM$_{2.5}$ by single particle aerosol mass spectrometry, Sci. Total Environ., 593–594, http://dx.doi.org/10.1016/j.scitotenv.2017.03.099, 2017.

Zhao, J., Du, W., Zhang, Y. J., Wang, Q. Q., Chen, C., Xu, W. Q., Han, T. T., Wang, Y. Y., Fu, P. Q., Wang, Z. F., Li, Z. Q., and Sun, Y. L.: Insights into aerosol chemistry during the 2015 China Victory Day parade: results from simultaneous measurements at ground level and 260 m in Beijing, Atmos. Chem. Phys., 17, 3215–3232, https://doi.org/10.5194/acp-17-3215-2017, 2017.

4. Since the rich-K type particles are often the dominant particle type, I am surprised that there isn't more time dedicated to describing what these particles might be. While this reviewer understands that these are clusters, the other clusters have much more intuitive names that can be traced back to certain sources. Similarly, I am surprised looking at Figure S3 that the all of the clusters have a large potassium peak. Is this usual for this SPAMS instrument?

**Response:** As the description shown in Supplementary Material (Text S1. Characteristics of particle composition), the particles containing the strongest potassium (*m/z* $^{39}$K$^+$) signal in the positive MS and significant sulfate (*m/z* $^{97}$HSO$_4^-$), and nitrate (*m/z* $^{46}$NO$_2^-$, $^{62}$NO$_3^-$) fragments in the negative MS are identified as Potassium-rich (rich-K) (Fig. S3a). The sources of the rich-K particles are complex, including biomass burning (Pratt et al., 2011), secondary formation (Bi et al., 2011; Shen et al., 2017), and industrial and traffic emissions (Zhang et al., 2017). A weak phosphate (*m/z* $^{79}$PO$_3^-$) signal is seen in Fig. S3a, consistent with the results of a study that $^{79}$PO$_3^-$ could be originated from motor vehicle lubricants (Yang et al., 2017). Significant peaks $^{97}$HSO$_4^-$ and $^{62}$NO$_3^-$ indicate that the rich-K particle might experience atmospheric aging after the primary biomass burning emission.

As the reviewer noted, many studies reported that SPAMS is very sensitive to alkali metal cations, and the ionization efficiencies of different chemical species are greatly varied (Gross et al., 2000; Xu et al., 2017). It should be noted that potassium with the desorption lasers used in the SPAMS could lead to its appearance in most particle types (Healy et al., 2013). SPAMS is extremely sensitive to potassium ($^{39}$K$^+$), thus its detection is possible even presents at a trace level such as in the exhausts of diesel- or bio-diesel-fueled vehicles (Giorio et al., 2015). Moreover, the peak at *m/z* 39 might not only be for potassium ([K]$^+$ and also for fragment [C$_3$H$_3$]$^+$ (Silva and Prather, 2000). The absence of common peaks associated with potassium (i.e. *m/z* $^{113}$ [K$_2$Cl]$^+$ and *m/z* $^{213}$ [K$_3$SO$_4$]$^+$), additionally of the ratio between *m/z* 39 and 41 is ~18 (the isotopic ratio for $^{39}$K/$^{41}$K is 13.28), further suggest that peak at *m/z* 39 is not for potassium only, whereas it generally dominates in biomass burning particles and other potassium-containing clusters (Silva et al., 1999; Dall'Osto et al., 2012). The presences of potassium and its cluster ions with chloride and sulfate allow the definitive identification of particles derived from biomass burning, sea salt, or soil particles. The combinations of both characteristic features allow for the comprehensive identification of biomass-burning particles, isolating them from other combustion-related particles, such as tobacco smoke. Moreover, the peaks at *m/z* 45,

59, and 71 are distinctive in the biomass-burning particles. The related elaborations in Supplementary materials (Text S1) have been revised as follows.

Line 16-28 in Supplementary materials (Text S1):

*"Nine particle groups are identified based on their chemical characteristics shown in the publications. General mass spectral characteristics for each particle group are presented in Fig. S3. The intense potassium ($^{39}K^+$) peak in almost all particles is attributed to the highly sensitive to potassium with the desorption laser used in the SPAMS (Gross et al., 2000; Xu et al., 2017; Giorio et al., 2015). Studies have reported that potassium by itself isn't an adequate marker for biomass burning because it presents in mass spectra of a variety of particle types (Healy et al., 2013). Moreover, the peak at m/z 39 might not be for $K^+$ and also for organic fragment $C_3H_3^+$ (Silva and Prather, 2000). Especially, the presence of other potassium clusters (i.e., m/z $^{113}K_2Cl^+$ or m/z $^{213}K_3SO_4^+$) and peaks at m/z 45, 59 and 71 are distinctive for biomass burning particles (Silva et al., 1999; Dall'Osto et al., 2012). The combination of the presence of phosphate (m/z $^{79}PO_3^-$) allows identification of particles derived from traffic emissions."*

**Reference:**

Dall'Osto, M., and Harrison, R. M.: Urban organic aerosols measured by single particle mass spectrometry in the megacity of London, Atmos. Chem. Phys., 12, 4127–4142, https://doi.org/10.5194/acp-12-4127-2012, 2012.

Gross, D. S., Gälli, M. E., Siliva, P. J., and Prather, K. A.: Relative sensitivity factors for alkali metal and ammonium cations in single-particle aerosol time-of-flight mass spectra, Anal. Chem., 72, 416–422, https://doi.org/10.1021/ac990434g, 2000.

Giorio, C., Tapparo, A., Dall'Osto, M., Beddows, D. C. S., Esser-Gietl, J. K., Healy, R. M., and Harrison, R. M.: Local and Regional Components of Aerosol in a Heavily Trafficked Street Canyon in Central London Derived from PMF and Cluster Analysis of Single-Particle ATOFMS Spectra, Environ. Sci. Technol., 49(6), 3330–3340, https://doi.org/10.1021/es506249z, 2015.

Silva, P. J., Liu, D.Y., Noble, C.A., and Prather, K. A.: Size and chemical characterization of individual particles resulting from biomass burning of local Southern California species, Environ. Sci. Technol., 33, 3068–3076, https://doi.org/10.1021/es980544p, 1999.

Silva, P. J. and Prather K. A.: Interpretation of mass spectra from organic compounds in aerosol time-of-flight mass spectrometry, Anal. Chem., 72, 3553–3562, https://doi.org/10.1021/ac9910132, 2000.

Healy, R. M., Sciare, J., Poulain, L., Crippa, M., Wiedensohler, A., Prévôt, A.S.H., Baltensperger, U., Sarda-Estève, R., McGuire, M. L., Jeong, C. H., McGillicuddy, E., O'Connor, I. P., Sodeau, J. R., Evans, G. J., and Wenger, J. C.: Quantitative determination of carbonaceous particle mixing state in Paris using single-particle mass spectrometer and aerosol mass spectrometer measurements, Atmos. Chem. Phys., 13, 9479–9496, http://dx.doi.org/10.5194/acp-13-9479-2013, 2013.

Xu, J., Li, M., Shi, G., Wang, H., Ma, X., Wu, J., Shi, X., and Feng, Y.: Mass spectra features of biomass burning boiler and coal burning boiler emitted particles by single particle aerosol

mass spectrometer, Sci. Total Environ., 598, 341–352, https://doi.org/10.1016/j.scitotenv.2017.04.132, 2017.

5. Figure 2 would be greatly improved if (a) the table was not inset into the figure, and (b) that the non-Chinese countries were not plotted in blue. For part (a), it is difficult to read the text; for part (b), blue colors on maps typically denote bodies of water.

**Response:** Suggestion taken. The original table inserted in Figure 2 was removed, and the table inside was moved to Supplementary (Table S1). In addition, Figure 2 with the embedded chart has been redrawn as below.

*"Table S1. Number concentration and relative fraction of the six main particle types in four trajectory clusters during the whole observation."*

| Type | Cluster 1 | | Cluster 2 | | Cluster 3 | | Cluster 4 | |
|---|---|---|---|---|---|---|---|---|
| | Counts | Fraction (%) | Counts | Fraction (%) | Counts | Fraction (%) | Counts | Fraction (%) |
| rich-K | 103275 | 32.7 | 1486 | 30.8 | 33319 | 26.8 | 4291 | 25.2 |
| BB | 58379 | 18.5 | 1731 | 35.9 | 23970 | 19.3 | 4562 | 26.8 |
| OC | 37856 | 12.0 | 744 | 15.4 | 19227 | 15.5 | 2138 | 12.5 |
| Ammonium | 39388 | 12.5 | 240 | 5.0 | 12948 | 10.4 | 1304 | 7.7 |
| EC-aged | 35143 | 11.1 | 245 | 5.1 | 9553 | 7.7 | 1077 | 6.3 |
| Dust | 28002 | 8.9 | 176 | 3.6 | 20613 | 16.6 | 2827 | 16.6 |

[Figure]

Figure 2. Maps of the mean HYSPLIT back trajectory clusters (72 h) at the height of 500 m during the whole field observation. Embedded pie chart represents the relative fraction of each particle type in the four clusters.

6. How are the errors bars generated for the particle types in each cluster in Figure 2?

**Response:** Figure 2 has been modified with the Comment 5. In original manuscript, the counts in Figure 2 are calculated by the average of the number concentration of each particle type during the total sampling periods for each cluster. In the revised manuscript, the counts have been changed to the ratio of the sum number concentration of each particle type to the sum of the total particle number in each cluster. The error bars are thus removed. The update of the relevant values shown in Table S1 with the Comment 5.

7. It might be useful to put the y-axis on Figure 3a, Figure 5a, and Figure 5b on a log scale. Mainly, I was interested in seeing how many particles there are of each type at the larger sizes where the number fraction plots get a little noisy.

**Response:** The new Figure R4a (original Figure 3a) put the y-axis on a log scale. There are relatively fewer particles with larger sizes for each particle type. For example, most count numbers are <100 at the size >1.2 μm, except for Dust. Especially, in new Fig. R4b and c, the particles with sizes >1.2 μm could not be seen due to the extremely small count number during Episodes 1 and 2.

Studies have reported that the SPAMS instrument is more sensitive to relatively small size particles. The higher sensitivity to chemical species with smaller particle sizes is attributed to (1) a greater volume fraction of small particles being vaporized by the ablation/ionization laser and (2) a lower probability of positive-negative charge recombination in the ablation plume for smaller-size particles (Bhave et al., 2002; Noble and Prather, 2000).

[Figure]

Figure R4. Size distributions of the particle count for nine particle types during the entire sampling campaign (a), and for major particle types during Episode 1 (b) and Episode 2 (c).

The particles with relatively lower concentrations also show a minor influence on the overview of size distribution. Since the size distributions of single particles from SPAMS are not scaled by other instruments such as SMPS, we have reduced the descriptions of the particle sizes in Section 3.2. In addition, as the count numbers with larger particle sizes fluctuate greatly after using a log scale of the y-axis, it might be reasonable and aesthetic to keep the y-axis intact.

**Reference:**

Bhave, P. V., Allen, J. O., Morrical, B. D., Fergenson, D. P., Cass, G. R., and Prather, K. A.: A Field-Based Approach for Determining ATOFMS Instrument Sensitivities to Ammonium and Nitrate, Environ. Sci. Technol., 36, 4868–4879, https://doi.org/10.5194/acp-12-4127-2012, 2002.

Noble, C. A., and Prather, K. A.: Real-time single particle mass spectrometry: A historical review of a quarter century of the chemical analysis of aerosols, Mass Spec. Rev., 19, 248–274, https://doi.org/10.1002/1098-2787(200007)19:4<248::AID-MAS3>3.0.CO;2-I, 2000.

8. Figure 7 suggests that knowing the oxidant concentration is not enough a priori knowledge to know the number fraction of particles that contain markers for secondary aerosol. Thus, you are missing some dimension that can make this analysis useful. Potentially, you are in an oxidant limited regime sometimes, and a precursor limited regime in other times. Perhaps also peak heights would be more useful than fraction containing, and could help tighten up the relationships.

**Response:** As shown in Fig. R5, the peak areas of each secondary ion ($^{43}C_2H_3O^+$, $^{89}HC_2O_4^-$, $^{18}NH_4^+$, $^{62}NO_3^-$, and $^{97}HSO_4^-$) are positively correlated with the concentrations of oxidant ($O_x$) during E2. However, there is no uniform linear correlation during E1.

[Figure]

Figure R5. Correlations between the peak area (PA) for five secondary ions and $O_x$ concentration during Episodes 1 and 2. The k and M represents thousands and ten thousands in the y-axis, respectively.

SPAMS has been made in quantifying individual chemical species either through multivariate analysis or by applying peak intensities for specific ions (e.g., Healy et al., 2013; Jeong et al., 2011; Xing et al., 2011). However, the ionization efficiency of different chemical species is difficult due to the variations in their ionization energies (Gross et al., 2000; Xu et al., 2017). Therefore, compared to absolute count number and peak area, number fraction, or relative peak area (RPA) are commonly applied because they are less sensitive to the variability in ion intensities associated with particle-laser interactions (Gross et al., 2000; Zhang et al., 2014; Zhang et al., 2020). Specifically, highly sensitive ions during the ionization could lead to a relatively higher peak area, and thus lower the RPA of other ion peaks.

RPA, defined as the peak area of each *m/z* divided by the total dual ion mass spectral peak area, is related to the relative amount of a species on a particle. To avoid the situation where the high peak area of some ions with high ionization efficiency (e.g., $^{39}K^+$) reduces the RPA of other ions, we tend to select the number fraction of $^{43}C_2H_3O^+$, $^{89}HC_2O_4^-$, $^{62}NO_3^-$, $^{97}HSO_4^-$, and $^{18}NH_4^+$-containing particles to investigate secondary formation mechanism.

**Reference:**

Gross, D. S., Gälli, M. E., Siliva, P. J., and Prather, K. A.: Relative sensitivity factors for alkali metal and ammonium cations in single-particle aerosol time-of-flight mass spectra, Anal. Chem., 72, 416–422, https://doi.org/10.1021/ac990434g, 2000.

Healy, R. M., Sciare, J., Poulain, L., Crippa, M., Wiedensohler, A., Prévôt, A.S.H., Baltensperger, U., Sarda-Estève, R., McGuire, M. L., Jeong, C. H., McGillicuddy, E., O'Connor, I. P., Sodeau, J. R., Evans, G. J., and Wenger, J. C.: Quantitative determination of carbonaceous particle mixing state in Paris using single-particle mass spectrometer and aerosol mass spectrometer measurements, Atmos. Chem. Phys., 13, 9479–9496, http://dx.doi.org/10.5194/acp-13-9479-2013, 2013.

Jeong, C. H., McGuire, M. L., Godri, K. J., Slowik, J. G., Rehbein, P.J.G., and Evans, G.J.: Quantification of aerosol chemical composition using continuous single particle measurements. Atmos. Chem. Phys., 11, 7027–7044, http://dx.doi.org/10.5194/acp-11-7027-2011, 2011.

Liang, Z. C., Zhou, L. Y., Infante C.R.A., Li, X. Y., Cheng, C. L., Li, M., Tang, R. Z., Zhang, R. F., Lee, P.K.H., Lai, A.C.K., and Chan, C. K.: Sulfate Formation in Incense Burning Particles: A Single-Particle Mass Spectrometric Study, Environ. Sci. Technol. Lett., 9, 718–725, https://doi.org/10.1021/acs.estlett.2c00492, 2022.

Xing, J. H., Takahashi, K., Yabushita, A., Kinugawa, T., Nakayama, T., Matsumi, Y., Tonokura, K., Takami, A., Imamura, T., Sato, K., Kawasaki, M., Hikida, T., Shimono, A.: Characterization of aerosol particles in the Tokyo Metropolitan area using two different particle mass spectrometers. Aerosol Science and Technology 45, 315–326, http://dx.doi.org/10.1080/02786826.2010.533720, 2011.

Xu, J., Li, M., Shi, G. L., Wang, H. T., Ma, X., Wu, J. H., Shi, X. R., and Feng, Y. C.: Mass spectra features of biomass burning boiler and coal burning boiler emitted particles by single particle aerosol mass spectrometer, Sci. Total Environ., 598, 341–352, https://doi.org/10.1016/j.scitotenv.2017.04.132, 2017.

Zhang, G. H., Lian, X. F., Fu, Y. Z., Lin, Q. H., Li, L., Song, W., Wang, Z. Y., Tang, M. J., Chen, D. H., Bi, X. C., Wang, X. M., and Sheng, G. Y.: High secondary formation of nitrogen-containing organics (NOCs) and its possible link to oxidized organics and ammonium, Atmos. Chem. Phys., 20, 1469–1481, https://doi.org/10.5194/acp-20-1469-2020, 2020.

9. Figure 8 is interesting because the trends are similar between the episodes; however, that they don't line up makes the interpretation difficult. Again, it seems that you're missing some dimension here that could help your

analysis—perhaps some smarter filtering by particle type could help? As it stands, the interpretation is muddled.

**Response:** As shown in Fig. R6, higher RH is seen RH during E1 (ranging from 37 to 99%) than E2 (25-77%). The aqueous-phase formation pathways of secondary species could be represented by the correlations between the number fraction of each secondary species and RH during nighttime (from 20:00 to 06:00 the next day) in the two episodes. During the nighttime, RH ranges from 79 to 99% in E1, and 43 to 77% in E2. Therefore, Fig. 7 is not suitable to be a line-up.

[Figure]

Figure R6. Time-series plots of PBL height, wind speed, relative humidity (RH), and the number concentration of the main six particle groups. Yellow shades correspond to Episodes 1 and 2, respectively.

It should be noted that it is still quite challenging for SPAMS to provide quantitative information on chemical compositions. Despite this, the number fraction and relative peak area (RPA) are still reliable indicators for investigating the atmospheric processing of various species in individual particles. Although they are insufficient to provide a quantitative assessment of secondary formation, our results successfully provide evidence for the possible pathways of secondary production in the southeastern margin of TP and, thus, can help achieve a qualitative understanding of the formation and evolution of secondary species in individual particles (Lian et al., 2021).

In addition, the reason for choosing the number fraction over the peak area in the presentation is provided in Comment 8. The correlation between peak area and RH also show in Fig. R7.

[Figure]

Figure R7. Correlations between the peak area (PA) for five secondary ions and RH during Episodes 1 and 2.

As per the reviewer's comment, Fig. R8 and R9 have been created. The number fractions of secondary ions are obtained by taking the percentage of the number concentration of each secondary ions in the whole detected particles. Moreover, the fraction of each particle type is also given by the ratio of the number concentration of each particle type to the whole detected particle. The mixing states are calculated by the ratio of the number concentration of secondary ions to each particle type. Therefore, both Fig. R8 and R9 could present their mixing states.

[Figure]

Figure R8. Correlation between the number fraction of the six particle types and $O_x$ concentration during Episodes 1 and 2.

[Figure]

Figure R9. Correlations between the number fraction of the six particle types and RH during Episodes 1 and 2.

As discussed in Comment 8, we tend to select the number fraction. A more detailed interpretation of the aqueous-phase reaction of secondary species in TP has been clarified as follows.

Line 406-440:

"*Fig. 7 illustrates that the number fractions of $^{43}C_2H_3O^+$, $^{89}HC_2O_4^-$, $^{97}HSO_4^-$, and $^{18}NH_4^+$ have moderate to strong positive correlations with RH (r = 0.70~0.81, p < 0.01 or 0.05) in the nighttime during the two episodes, except that $^{43}C_2H_3O^+$ during E2 (p = 0.48) and $^{89}HC_2O_4^-$ during E1 (p = 0.12). Furthermore, $^{62}NO_3^-$ fraction has no obvious changes with insignificant correlation with RH during E1 (p = 0.43) and presents a moderate negative correlation with RH (r = 0.69, p < 0.01) during E2. As shown in Fig. 7e, the highest aqueous formation rate of $HSO_4^-$ is mainly due to the properties of low volatile and high hygroscopic sulfate (Wang et al., 2016; Zhang et al., 2019c; Sun et al., 2013). Compared with that during E2 (slop=0.014), the decreased formation rate of $HSO_4^-$ during the E1 (slop=0.009) may be because of the decreases of aerosol acidity in higher RH > 80% (Huang et al., 2019; Meng et al., 2014; Tian et al., 2021). And the increased contributions of regional transport due to the high WS (2.4 ± 0.8 m s$^{-1}$) during E2 are comparable to the low WS (0.08 ± 0.08 m s$^{-1}$) during E1 (Fig. S8). The fair production rate of $NH_4^\pm$ during the E1 (slop=0.005) and E2 (slop=0.006) demonstrate that an aqueous-phase reaction could effectively promote ammonium formation. Meanwhile, a slightly larger slop during E2 could be also affected by the increased contributions of regional transport. Compared with those during E1, the inverse generation rates of two secondary organic species (i.e., $C_2H_3O^+$ and $HC_2O_4^-$) during E2 are possibly caused by the different formation pathways with a variety of RH levels or distinct regional transports. For example, $C_2H_3O^+$ shows a strong correlation with RH (r = 0.70, p < 0.05) during E1 (slop=0.003)*

*but has an insignificant correlation during E2. This could be explained by high RHs that could effectively promote secondary organic formation during E1. In addition, the $HC_2O_4^-$ fraction increases slightly (9.7-13.1%) during E1 is potentially ascribed to more abundant Dust-type particles (20.3%) which compose of high calcium (Ca) (Fig. S13) that favor the formation of metal oxalate complexes (i.e., Ca oxalate). At high RHs (93.4 ± 7.6%), if oxalate ions are dissolved in the aqueous phase with the presence of Ca ions, the Ca oxalate complexes can precipitate because of their low hygroscopic and insoluble natures (Furukawa and Takahashi, 2011). This could offset the oxalate formation in the aqueous-phase reaction. However, significant linear increases (slop is 0.003) with RH (r = 0.81, p < 0.01) during E2 demonstrate that the aqueous-phase reaction effectively promotes the oxalate formation (Cheng et al., 2017; Meng et al., 2020). No obvious change and insignificant correlation between $^{62}NO_3^-$ and RH are found during E1, potentially attributed to the decreases of $NO_2$ concentration (3.7 ± 0.4 ppbv) in the local atmosphere. Meanwhile, high RHs could promote organonitrates formation (Fang et al., 2021; Fry et al., 2014). The linearity between $^{62}NO_3^-$ and RH (r = 0.69, p < 0.01) significantly decreases during E2, mostly due to the losses of the volatile compound through the regional transport (Fig. S14)."*

[Figure]

Figure S13. Correlations between the relative fraction of oxalate and Dust type during E1.

[Figure]

Figure S14. Correlations between the relative fraction of nitrate (62NO3−) and wind speed during E2.

**Reference:**

Furukawa, T., and Takahashi, Y.: Oxalate metal complexes in aerosol particles: implications for the hygroscopicity of oxalate-containing particles. Atmos. Chem. Phys., 11, 4289–4301, https://doi.org10.5194/acp-11-4289-2011, 2011.

Huang, X. J., Zhang, J. K., Luo, B., Luo, J. Q., Zhang, W., and Rao, Z. H.: Characterization of oxalic acid-containing particles in summer and winter seasons in Chengdu. China, Atmos. Environ., 198, 133–141. https://doi.org/10.1016/j.atmosenv.2018.10.050, 2019.

Lian, X. F., Zhang, G. H., Yang, Y. X., Lin, Q. H., Fu, Y. Z., Jiang, F., Peng, L., Hu, X. D., Chen, D. H., Wang, X. M., Peng, P. A., Sheng, G. Y., and Bi, X. H.: Evidence for the Formation of Imidazole from Carbonyls and Reduced Nitrogen Species at the Individual Particle Level in the Ambient Atmosphere. China, Environ. Sci. Technol. Lett., 8, 9−15. https://dx.doi.org/10.1021/acs.estlett.0c00722, 2021.

---

## Author Comment (AC4)

**Responses to Referee #2:**

The response by Li et al. addressed this reviewer's comments in a point-by-point manner. While the authors had adequately addressed many of my comments, this reviewer feels that the authors have not fully addressed several main comments. To increase the clarity of the paper, and the validity of the results, I suggest that the authors addressed the comments below.

**Response:** We highly appreciate the valuable suggestions by the reviewer, which are helpful for us to improve the quality of our manuscript. We have carefully addressed the comments in point-by-point form as shown below. Detailed responses to the comment are highlighted in blue, and the revised text is *underlined in italics*. Attached please also find the marked-up manuscript with tracked changes in the revised manuscript.

**2** General Comments**

• **Reviewer Point 3:** I suggest that the authors start their conversation about the size distributions on Line 270, with their caveat that starts on Line 293. As it stands, the authors discuss the size distributions as if the SPAMS size distribution is quantitative–but it largely depends on the SPAMS detection efficiency as a function of size. The authors do segue into a conversation about the number fractions as a function of size, which is much more valid. The authors slide back into using the SPAMS size distribution again starting on Line 285, which again, without a quantitative sizing instrument is hard to interpret. Finally, the authors reference Figure S10a and S10b, but the current supplemental does not contain any figures with size distributions.

**Response:** Suggestion taken. The statement has been revised as follows.

**Line 305-313**

"Compared with the total particle size distribution, the peak values of the six main particle types show minor differences (< 80 nm) during the two different episode periods (Fig. 11b,c). However, the percentage of the six particle types distribute in wider size ranges during E2 than during E1 possibly due to the more intensive atmospheric aging. Similarly, during the two episodes (Fig. 3b,c), the relatively high fraction of the rich-K and BB particles are more affected by the primary emissions when their peak value concentrate at < 300 nm, and > 300 nm are more related to the aging process (Li et al., 2022b; Bi et al., 2011). Relatively greater fluctuation for the large-size fractions (> 1.1  $\mu$ m) could be explained by the low particle concentration (a number less than 20)."

Additionally, sorry for the confusion that the previous supplemental information possibly uploads unsuccessfully. I will upload all materials more carefully this time.

**The previous versions of Figure S11 (a,b) are shown in Figure R1 below.**

Figure R1. Size distributions of the (a, c) number concentrations and (b, d) fractions of the major six particle types (rich-K, BB, OC, Ammonium, EC-aged, and Dust) during two episodes of (a, b) E1 and (c, d) E2.

Under the reviewer's suggestion, in order to more clearly illustrate the characteristics of particle size distribution, we have replaced the figures in the current main text (Figure 3 in follows) and supplemental (Figure S10 in follows) respectively, because the number fraction is more valid.

---

## Referee Report (RR1)

**Anonymous Re-Review of *In-depth study of the formation processes of single atmospheric particles in the southeastern margin of Tibetan Plateau**

Anonymous Reviewer

May 2023

**1 Summary**

The response by Li *et al.* addressed this reviewer's comments in a point-by-point manner. While the authors had adequately addressed many of my comments, this reviewer feels that the authors have not fully addressed several main comments. To increase the clarity of the paper, and the validity of the results, I suggest that the authors addressed the comments below.

**2 General Comments**

- Reviewer Point 3: I suggest that the authors start their conversation about the size distributions on Line 270, with their caveat that starts on Line 293. As it stands, the authors discuss the size distributions as if the SPAMS size distribution is quantitative–but it largely depends on the SPAMS detection efficiency as a function of size. The authors do segue into a conversation about the number fractions as a function of size, which is much more valid. The authors slide back into using the SPAMS size distribution again starting on Line 285, which again, without a quantitative sizing instrument is hard to interpret. Finally, the authors reference Figure S10a and S10b, but the current supplemental does not contain any figures with size distributions.

- Reviewer Point 4: While the authors have pointed the reviewer to additional information about the rich-K particles in the supplemental material, I would like to see the authors add a description of the rich-K particles to the main text. As currently written, the text suggests that the rich-K are one type of particle from a particular source. The supplemental information, however, suggests that the particles are from different sources (biomass burning, traffic emissions, and secondary sources). Thus,

it seems like there is a disconnect in using rich-K particles in the back trajectory cluster analysis. If this one particle type has different sources, then it seems like their relative fractions in the back trajectories could have several causes. Thus, this reviewer suggests that the authors clarify why they use this one cluster for particles with several sources–perhaps they have one distinct source in that they're anthropogenic? Finally, this reviewer is also surprised that potassium aerosol is formed in secondary reactions. I think this needs further explanation in the main text.

- Reviewer Point 8/9: To this reviewer, the results in Figures R6 and R7 seem to contradict the results in Figures 6 and 7, respectively. Figure R6 suggests that higher $O_x$ concentrations lead to higher secondary-aerosol peaks. Similarly, in Figure R7, higher RH leads to higher secondary peaks. In some cases, the opposite conclusions are reached using Figures 6 and 7. Finally, the authors should clarify that these results are speculative, because the current $O_x$ and RH conditions are not an indicator of the past $O_x$ and RH conditions that a particle has experienced.

---

## Referee Report (RR2)

**Anonymous Re-Review of *In-depth study of the formation processes of single atmospheric particles in the southeastern margin of Tibetan Plateau**

Anonymous Reviewer

July 2023

**1    General Comment**

The response by Li *et al.* addressed this reviewer's second round of comments in a point-by-point manner. While the authors had adequately addressed almost all of these comments, it feels like the authors are still not being clear enough in section 3 that the "particle size distributions" are "SPAMS-specific size distributions." The authors also need to be clear that their discussion of the SPAMS specific size distributions, especially as they pertain to the interpretation of particle-type sources, is dependent on the detection efficiency of this particular SPAMS instrument.

---

## Author Response (AR2)

**Responses to Referee #2:**

The response by Li et al. addressed this reviewer's comments in a point-by-point manner. While the authors had adequately addressed many of my comments, this reviewer feels that the authors have not fully addressed several main comments. To increase the clarity of the paper, and the validity of the results, I suggest that the authors addressed the comments below.

**Response:** We highly appreciate the valuable suggestions by the reviewer, which are helpful for us to improve the quality of our manuscript. We have carefully addressed the comments in point-by-point form as shown below. Detailed responses to the comment are highlighted in blue, and the revised text is *underlined in italics*. Attached please also find the marked-up manuscript with tracked changes in the revised manuscript.

**2 General Comments**

• **Reviewer Point 3:** I suggest that the authors start their conversation about the size distributions on Line 270, with their caveat that starts on Line 293. As it stands, the authors discuss the size distributions as if the SPAMS size distribution is quantitative–but it largely depends on the SPAMS detection efficiency as a function of size. The authors do segue into a conversation about the number fractions as a function of size, which is much more valid. The authors slide back into using the SPAMS size distribution again starting on Line 285, which again, without a quantitative sizing instrument is hard to interpret. Finally, the authors reference Figure S10a and S10b, but the current supplemental does not contain any figures with size distributions.

**Response:** Suggestion taken. The statement has been revised as follows.

Line 305-313
*"Compared with the total particle size distribution, the peak values of the six main particle types show minor differences (< 80 nm) during the two different episode periods (Fig. 11b,c). However, the percentage of the six particle types distribute in wider size ranges during E2 than during E1 possibly due to the more intensive atmospheric aging. Similarly, during the two episodes (Fig. 3b,c), the relatively high fraction of the rich-K and BB particles are more affected by the primary emissions when their peak value concentrate at < 300 nm, and > 300 nm are more related to the aging process (Li et al., 2022b; Bi et al., 2011). Relatively greater fluctuation for the large-size fractions (> 1.1 μm) could be explained by the low particle concentration (a number less than 20)."*

Additionally, sorry for the confusion that the previous supplemental information possibly uploads unsuccessfully. I will upload all materials more carefully this time.

The previous versions of Figure S11 (a,b) are shown in Figure R1 below.

[Figure]

Figure R1. Size distributions of the (a, c) number concentrations and (b, d) fractions of the major six particle types (rich-K, BB, OC, Ammonium, EC-aged, and Dust) during two episodes of (a, b) E1 and (c, d) E2.

Under the reviewer's suggestion, in order to more clearly illustrate the characteristics of particle size distribution, we have replaced the figures in the current main text (Figure 3 in follows) and supplemental (Figure S10 in follows) respectively, because the number fraction is more valid.

[Figure]

Figure 3. Size distributions of the relative number fraction (%) of the total particles for nine groups during (a) the total sampling campaign and two episodes of (b) E1 and (c) E2.

[Figure]

Figure S11. Size distributions of the number concentrations of the nine particle types during (a) the total observation periods and two episodes of (b) E1 and (c) E2.

• **Reviewer Point 4:** While the authors have pointed the reviewer to additional information about the rich-K particles in the supplemental material, I would like to see the authors add a description of the rich-K particles to the main text. As currently written, the text suggests that the rich-K are one type of particle from a particular source. The supplemental information, however, suggests that the particles are from different sources (biomass burning, traffic emissions, and secondary sources). Thus, it seems like there is a disconnect in using rich-K particles in the back trajectory cluster analysis. If this one particle type has different sources, then it seems like their relative fractions in the back trajectories could have several causes. Thus, this reviewer suggests that the authors clarify why they use this one cluster for particles with several sources–perhaps they have one distinct source in that they're anthropogenic? Finally, this reviewer is also surprised that potassium aerosol is formed in secondary reactions. I think this needs further explanation in the main text.

**Response:** Previous studies have shown that the source of rich-K is complex, which includes biomass burning, secondary formation, industrial emission, and traffic emission (e.g., Pratt et al., 2011; Bi et al., 2011; Shen et al., 2017; Zhang et al., 2017). According to the characteristics of mass spectrum (e.g., the existence of $^{79}PO_3^-$ signal indicates the sources of traffic emission) and diurnal variation (e.g., similar with BB), we point out that traffic emission and biomass burning are contributed to the chemical component of rich-K type. The coexistence strong signals of sulfate ($m/z$ $^{97}HSO_4^-$) and nitrate ($m/z$ $^{46}NO_2^-$ and $^{62}NO_3^-$) in the negative mass spectrum indicate that rich-K particles have undergone a certain degree of atmospheric process or affected by secondary formation. By analyzing the correlation between seven variables (Fig. S4), rich-K type is strongly correlated with Ammonium (r=0.84) and EC-aged (r=0.90) types, then well correlated with OC (r=0.70) and BB (r=0.68) types. These results further demonstrate that rich-K particle is sourced from traffic emission and biomass burning, and affected by secondary formation during the atmospheric aging.

Because the SPAMS itself is sensitive to potassium ($^{39}K^+$), in some cases, the $^{39}K^+$ signal has weak indicative. Therefore, other components (except for $^{39}K^+$) are usually combined to indicate the potential sources of specific particles. For example,

in addition to $^{39}K^+$, the biomass burning emissions also need to contain levoglucosan fragments ($^{45}CHO_2^-$, $^{59}C_2H_3O_2^-$, $^{71}C_3H_3O_2^-$, $^{73}C_3HO_3^-$) and $^{113,115}K_2Cl^+$ signals, etc., so as to further prove its accurate source. It should be emphasized that secondary formation only means that the particle type contains the abundance of sulfate and nitrate signals in its negative mass spectrum, and has no other obvious components except potassium.

[Figure]

Figure S4. The correlation results between seven variables ($p < 0.01$) was statistically analyzed by IBM SPSS software (version 23). The values in the figure represent Pearson's $r$.

According to the reviewer's suggestion, we have added the following explanations to the main text.

Line 190-191
*"Their characteristics of mass spectrum and possible sources are described in supplemental information of text S1 in detail."*

Line 193-202
*"Combined with the previous studies and the characteristics of the mass spectrum (Fig. S3) in this study, the rich-K particles are contributed by biomass burning and traffic emission, because that extensive works usually identify abundant $^{39}K^+$ signal for biomass burning (Pratt et al., 2011; Chen et al., 2017), while the presence of phosphate (m/z $^{79}PO_3^-$) indicates the vehicle exhaust (Yang et al., 2017). The results of the correlation between seven variables (Fig. S4) show that rich-K type was strongly correlated with Ammonium (r=0.84) and EC-aged (r=0.90) types, follow well correlated with OC (r=0.70) and BB (r=0.68) types, further demonstrate that rich-K particles type is from traffic emission and biomass burning, and is affected by secondary formation during the atmospheric aging in southeastern TP."*

Line 243-250

*"Cluster 3 and 4 have the comparable contributions of OC (15.5% and 12.5%, respectively), increased of BB (19.3% and 26.8%, respectively) and decreased of rich-K (26.8% and 25.2%, respectively), Ammonium (10.4% and 7.7%, respectively) and EC-aged (7.7% and 6.3%, respectively), to those of Cluster 1, but with a high contribution of Dust (16.6%), which refer Cluster 3 and 4 to as dust and biomass burning pollution. However, Cluster 1 is more influenced by compound pollution, mainly including secondary formation, biomass burning, and traffic emissions."*

Line 253-254
*"A stable diurnal variation of rich-K fraction is mainly due to its large proportion and diverse sources."*

**Reference:**

Chen, Y., Wenger, J. C., Yang, F. M., Cao, J. J., Huang, R. J., Shi, G. M., Zhang, S. M., Tian, M., and Wang, H. B.: Source characterization of urban particles from meat smoking activities in Chongqing, China using single particle aerosol mass spectrometry, Environ. Pollut., 228, 92–101, https://doi.org/10.1016/j.envpol.2017.05.022, 2017.

Pratt, K. A., Murphy, S. M., Subramanian, R., DeMott, P. J., Kok, G. L., Campos, T., Rogers, D. C., Prenni, A. J., Heymsfield, A. J., Seinfeld, J. H., and Prather, K. A.: Flight-based chemical characterization of biomass burning aerosols within two prescribed burn smoke plumes, Atmos. Chem. Phys., 11, 12549–12565, https://doi.org/10.5194/acp-11-12549-2011, 2011.

• **Reviewer Point 8/9:** To this reviewer, the results in Figures R6 and R7 seem to contradict the results in Figures 6 and 7, respectively. Figure R6 suggests that higher Ox concentrations lead to higher secondary-aerosol peaks. Similarly, in Figure R7, higher RH leads to higher secondary peaks. In some cases, the opposite conclusions are reached using Figures 6 and 7. Finally, the authors should clarify that these results are speculative, because the current Ox and RH conditions are not an indicator of the past Ox and RH conditions that a particle has experienced.

**Response:** As the reviewer said, Figures R2 and R3 suggest that higher Ox concentrations and RH values lead to higher peak areas of secondary species. Generally, the increase in secondary component concentration is accompanied by the increase in total particle concentration. Thus, it should be explained that the increase in the total particles may be affected by the primary emission and meteorological conditions (such as regional transport) (Fig. S10 and S13). The increased trend of the ratio of each secondary species to the total particles effectively indicates that the secondary species have more formation. For all that, the current results and interpretations are speculative, because ① SPAMS only be qualitatively analyzed the individual particles information rather than quantitatively interpreted due to the limited detection rate; ② the current $O_x$ and RH conditions are not an indicator of the past $O_x$ and RH conditions that a particle has experienced. After taking the reviewer's suggestion, the statement has been revised as follows.

Line 385-392

*"The oxidant $O_x$ ($O_3$ + $NO_2$) concentration and RH usually serve as indicators of the degree of photochemical oxidation (Wood et al., 2010) and aqueous-phase reaction (Ervens et al., 2011), receptively, though the current $O_x$ and RH conditions obtained using the in-situ measurement are not indicative of the past conditions experienced by the particle. Thus, the relative number fractions of $^{43}C_2H_3O^+$, $^{89}HC_2O_4^-$, $^{62}NO_3^-$, $^{97}HSO_4^-$ and $^{18}NH_4^+$-containing particles to the total detected particles were selected to provide a rough speculative of the secondary formation mechanism in TP ambient conditions (Liang et al., 2022)."*

Line 491-497

*"Although the detailed formation pathways and their percentage contributions to secondary species are not quantitatively estimated in this study, our results have important implications for the various possibilities affecting the characteristic of chemical components, size distribution, mixing states, and formation mechanism of aerosols in the southeast TP. More depth investigations concerning the evolution mechanisms of secondary aerosols are encouraged since TP is a significant regulator to global climate change."*

[Figure]

Figure R2. Correlations between the peak area (PA) for five secondary ions and $O_x$ concentration during Episodes 1 and 2. The k and M represents thousands and ten thousands in the y-axis, respectively.

[Figure]

Figure R3. Correlations between the peak area (PA) for five secondary ions and RH during Episodes 1 and 2.

[Figure]

Figure S10. Correlations between oxidant ($O_x$) concentration and (a) PBL, (b) RH, (c) $NO_2$ concentration and (d) temperature during the Episode 1 (the open shape) and Episode 2 (the solid shape).

[Figure]

Figure S13. Correlations between RH and (a) PBL height, (b) O₃ concentration, (c) NO2 concentration and (d) temperature during the Episode 1 (the open shape) and Episode 2 (the solid shape).

---

## Author Response (AR3)

**Responses to Referee:**

**1 General Comment**

The response by Li et al. addressed this reviewer's second round of comments in a point-by-point manner. While the authors had adequately addressed almost all of these comments, it feels like the authors are still not being clear enough in section 3 that the "particle size distributions" are "SPAMS-specific size distributions." The authors also need to be clear that their discussion of the SPAMS specific size distributions, especially as they pertain to the interpretation of particle-type sources, is dependent on the detection efficiency of this particular SPAMS instrument.

**Response:** Following the reviewer's suggestion, we have added the following explanations in Section "2. Methodology".

Line 142-148

*"Particles measured by SPAMS mostly are within the size range of vacuum aerodynamic diameter ($d_{va}$) 0.2-2.0 µm.* *This SPAMS-specific size distribution is semi-quantitative evaluated the relative concentration and contribution of each particle type, mainly due to it largely dependence on the particle-detection efficiency (Allen et al., 2000; Yang et al., 2017). The characteristics of SPAMS-specific size distribution are statistical results, while the comparison of the relative distribution and number fraction of different particle types in each size bin are significant."*

In order to more clearly emphasize the specific characteristics of particle size distribution using SPAMS, we have replaced the related "particle size distributions" as "SPAMS-specific size distributions" in the current main text (in Section 3.2) and figure captions. Detailed revision also finds the marked-up manuscript with tracked changes in the revised manuscript.

**Reference:**

Allen, J. O., Fergenson, D. P., Gard, E. E., Hughes, L. S., Morrical, B. D., Kleeman, M. J., Gross, D. S., Galli, M. E., Prather, K. A., and Cass, G. R.: Particle Detection Efficiencies of Aerosol Time of Flight Mass Spectrometers under Ambient Sampling Conditions, Environ. Sci. Technol., 34, 211–217, http://dx.doi.org/10.1021/es9904179, 2000.

Yang, J., Ma, S. X., Gao, B., Li, X. Y., Zhang, Y. J., Cai, J., Li, M., Yao, L. A., Huang, B., and Zheng, M.: Single particle mass spectral signatures from vehicle exhaust particles and the source apportionment of on-line PM2.5 by single particle aerosol mass spectrometry, Sci. Total Environ., 593–594, http://dx.doi.org/10.1016/j.scitotenv.2017.03.099, 2017.